# A spatially resolved brain region- and cell type-specific isoform atlas of the postnatal mouse brain

Anoushka Joglekar [1], Andrey Prjibelski[2], Ahmed Mahfouz[3,4,5], Paul Collier[1], Susan Lin [6,7], Anna Katharina Schlusche[1], Jordan Marrocco [8], Stephen R. Williams[9], Bettina Haase[10], Ashley Hayes[9], Jennifer G. Chew[9], Neil I. Weisenfeld[9], Man Ying Wong[11], Alexander N. Stein[12], Simon A. Hardwick[1,13], Toby Hunt[14], Qi Wang[15], Christoph Dieterich[15], Zachary Bent[9], Olivier Fedrigo[10], Steven A. Sloan[16], Davide Risso[17], Erich D. Jarvis[10,18], Paul Flicek [14], Wenjie Luo [11], Geoffrey S. Pitt [6,7], Adam Frankish[14], August B. Smit[19], M. Elizabeth Ross [1] & Hagen U. Tilgner [1✉]

Splicing varies across brain regions, but the single-cell resolution of regional variation is unclear. We present a single-cell investigation of differential isoform expression (DIE) between brain regions using single-cell long-read sequencing in mouse hippocampus and prefrontal cortex in 45 cell types at postnatal day 7 (www.isoformAtlas.com). Isoform tests for DIE show better performance than exon tests. We detect hundreds of DIE events traceable to cell types, often corresponding to functionally distinct protein isoforms. Mostly, one cell type is responsible for brain-region specific DIE. However, for fewer genes, multiple cell types influence DIE. Thus, regional identity can, although rarely, override cell-type specificity. Cell types indigenous to one anatomic structure display distinctive DIE, e.g. the choroid plexus epithelium manifests distinct transcription-start-site usage. Spatial transcriptomics and long-read sequencing yield a spatially resolved splicing map. Our methods quantify isoform expression with cell-type and spatial resolution and it contributes to further our understanding of how the brain integrates molecular and cellular complexity.

[1] Brain and Mind Research Institute and Center for Neurogenetics, Weill Cornell Medicine, New York, NY, USA. [2] Center for Algorithmic Biotechnology, Institute of Translational Biomedicine, St. Petersburg State University, St Petersburg, Russia. [3] Department of Human Genetics, Leiden University Medical Center, Leiden 2333 ZC, The Netherlands. [4] Leiden Computational Biology Center, Leiden University Medical Center, Leiden 2333 ZC, The Netherlands. [5] Delft Bioinformatics Lab, Delft University of Technology, Delft 2628 XE, The Netherlands. [6] Graduate Program in Neuroscience, Weill Cornell Medical College, 1300 York Avenue, New York, NY 10065, USA. [7] Cardiovascular Research Institute, Weill Cornell Medicine, New York, NY, USA. [8] Harold and Margaret Milliken Hatch Laboratory of Neuroendocrinology, The Rockefeller University, New York, NY, USA. [9] 10x Genomics, Pleasanton, CA, USA. [10] The Vertebrate Genomes Lab, The Rockefeller University, New York, NY, USA. [11] Brain and Mind Research Institute and Appel Alzheimer's Research Institute, Weill Cornell Medicine, New York, NY, USA. [12] School of General Studies, Columbia University, New York, NY, USA. [13] Genomics and Epigenetics Division, Garvan Institute of Medical Research, Sydney, NSW, Australia. [14] European Molecular Biology Laboratory, European Bioinformatics Institute, Hinxton, UK. [15] Section of Bioinformatics and Systems Cardiology, University Hospital, 96120 Heidelberg, Germany. [16] Department of Human Genetics, Emory University School of Medicine, Atlanta, GA, USA. [17] Department of Statistical Sciences, University of Padova, Padova, Italy. [18] Howard Hughes Medical Institute, Chevy Chase, MD, USA. [19] Department of Molecular and Cellular Neurobiology, Center for Neurogenomics and Cognitive Research, Amsterdam Neuroscience, VU University, Amsterdam, The Netherlands. ✉email: hut2006@med.cornell.edu

Alternative splicing (AS) affects almost all spliced genes in mammals[1,2], vastly expands the proteome[3] and increases functional diversity of cell types[4]. Alternative transcription start sites (TSS) and poly-adenylation (polyA) sites further expand the alternative isoform landscape, regulating development, differentiation, and disease[5–9]. These RNA variables often depend on each other[10–13], and how their combined status impacts individual molecules can only be assessed using long-read sequencing[11,12,14–17], which sequences transcripts in single reads with no assembly required, thereby reducing alternative transcript assembly errors and enabling accurate isoform quantification.

Brain AS is especially diverse[18] and brain-region specific expression patterns of splicing factors[19] and other RNA-binding proteins[20] drive brain-region-specific splicing. Examples include diseases implicated by genes such as *MAPT, Bin1,* and neurexins[16,17,21]. Brain-region-specific isoform expression can either originate from molecular regulation in one or multiple cell types, or can arise purely from gene-expression or cell-type abundance differences without splicing regulation. These distinct models are especially important during postnatal development. For instance, in hippocampus (HIPP) and pre-frontal cortex (PFC), multiple cell types undergo differentiation, which is influenced by development-specific splicing[1,22–25] distinct from that of mature cell types. However, no cell-type-specific isoform investigation across brain regions exists to-date, owing to limitations in technology, throughput, and testing methods. HIPP and PFC are highly specialized regions of the telencephalon, and their circuitry is heavily implicated in movement control, cognition, learning, and memory formation. Disorders involving HIPP and PFC manifest in cognitive deficits, and understanding changes occurring at crucial developmental timepoints of these structures is important for case–control studies. Here, we employ single-cell isoform RNA sequencing (ScISOrSeq)[26] with increased throughput in HIPP and PFC at mouse postnatal day 7 (P7) to test and define cell-type-specific contributions to brain-region-specific splicing. Furthermore, we devised a spatial isoform expression method, which provides a spatial exon expression map (see www.isoformAtlas.com) in addition to the existing spatial gene-expression map of the Allen developing brain atlas.

## Results

### Short read clustering of P7 hippocampus and prefrontal cortex tissue assigns precursors to known adult cell-types.
Our ScI-SOrSeq approach used barcoded single cells followed by both short and long-read analyses to reveal splice variants specific to cell types (Fig. 1a). We identified cell types first using single-cell 3′-end sequencing. Short-read clustering across two hippocampal replicates revealed no need for integration anchors[27] to correct for batch effects (Fig. S1a). Characteristic markers[28,29] for 24 clusters in HIPP (Fig. S1b) identified eight glial types, including two astrocyte, three oligodendrocyte, a radial glia like (RGLs), ciliated ependymal, and secretory choroid plexus epithelial (CPE) cell clusters. Furthermore, we observed six vascular and immune populations including vascular endothelial cells, microglia, and macrophages (Fig. 1b). RNA velocity analysis revealed neuronal lineages in various differentiation stages (Fig. S1c, d): a neuronal intermediate progenitor cell (NIPC) population; three dentate gyrus granule neuroblast clusters (DG-GranuleNB); and three clusters each of excitatory (EN) and inhibitory neurons (IN). Alignment of our P7 data with published P30 hippocampal data[30] revealed subtype identities (CA3, CA1, Subiculum) for three excitatory neuron clusters and medial ganglionic eminence (MGE) and non-MGE-derived interneurons[31–33] (*PV +, Sst +,*

*Lamp5 +, Vip +*) in one cluster (IN1), distinct from Cajal–Retzius (IN3) cells (Fig. 1b, Supplementary Fig. 1e–g).

Similar analysis in PFC revealed seven glial clusters including astrocytes, oligodendrocytes, six populations of vascular and immune cells, and seven neuronal types[34,35] with confirmation of intermediate states from RNA velocity (Fig. 1c, Fig. S2a–d). Alignment with public P30 cortex data[30] further subdivided neuronal clusters into known cortical excitatory and interneuron classes (Fig. S2e–g). In contrast to HIPP, the MGE (*PV +, Sst +*) and non-MGE interneurons (*Vip +, Lamp5 +*) in the PFC were better separated into two clusters (IN1, IN2), while Cajal–Retzius cells again clustered separately (IN3). We identified excitatory neurons corresponding to different cortical layers which are not well-differentiated at P7. P4 ISH images (Image credit: Allen Institute) alongside gene expression projected onto the UMAP plots further validated our cell-type identification for both regions (Fig. 1d).

### A gene-wise test to determine differential isoform expression (DIE).
We next conducted long-read sequencing on our single-cell full-length HIPP and PFC cDNA (Supplementary Table 1), and deconvolved reads for each cell type using single-cell barcodes (Fig. 1a) for two independent replicates (Fig. S3, SS3). Differential exon usage between two conditions has been successfully assessed using a $2 \times 2$ contingency table per exon[2]. Using this method on our long-read data from HIPP and PFC yielded 31 genes (1.45%, $n = 2132$) exhibiting differential exon usage after Benjamini–Yekutieli (BY) correction[36] for dependent tests (Fig. S5a). Given this harsh correction, we devised a more sensitive gene-level test that considers TSS and polyA-sites in addition to exon connectivity. In this test, we count isoforms per gene in both conditions, leading to a n x 2 table. This yields fewer and independent tests, allowing for the Benjamini–Hochberg (BH) correction[36] and reducing false negatives. In each brain region, we define "percent isoform" ($\Pi$) as an isoform's relative abundance among its gene's transcripts. Similarly to requiring a $\Delta\Psi \geq 0.1$ for short reads[2], we require FDR $< = 0.05$ and $\Delta\Pi \geq 0.1$. This $\Delta\Pi \geq 0.1$ can be contributed collectively by at most two isoforms, provided their isoform $\Delta\Pi$ point in the same direction, to consider a gene exhibiting differential isoform expression (DIE, Fig. 2a).

In contrast to the 31 significant genes derived by the exon-based test, 395 genes (FDR $<= 0.05$, $\Delta\Pi \geq 0.1$; 9.06%; $n = 3958$) exhibited DIE when comparing HIPP and PFC isoforms using the gene-level test (Fig. 2b). The multiple testing correction factor influenced the significant gene number. For example, the *H13* gene (Fig. S5b) had a p-value of $1.7 \times 10^{-4}$ (uncorrected) and $2.7 \times 10^{-3}$ (corrected) by isoform tests, with a Benjamini–Hochberg correction factor of 15.6. The same gene's alternative exon had a p-value of $1.3 \times 10^{-4}$ (uncorrected) and 0.057 (corrected, non-significant) by exon tests with a Bejamini–Yekutieli correction factor of 431.3. Thus, gene-wise isoform testing is more sensitive (Fig. 2c, d). Concordantly, the maximum $\Delta\Psi$ for genes with DIE is higher than genes without DIE ($p \leq 0.015$, Wilcoxon-rank-sum test) (Fig. 2e).

Since our gene-level test considers exon connectivity, we can identify the exact isoforms contributing to DIE. Among the top two contributing isoforms for each of the 395 genes exhibiting regional DIE, we identified 76 high-confidence novel isoforms (Methods). Manual validation using GENCODE criteria confirmed all 76. Functionally, 40 (52.6%) are coding transcripts, 24 (31.6%) show nonsense mediated decay (NMD), 11 (14.5%) show intron retention, and one isoform is of a long noncoding gene (*Meg3*) (Fig. 2f). Such non-coding and NMD transcripts indicate region-specific regulation[1]. To pinpoint the source of isoform differences in the 395 genes, we tested TSS and polyA-site

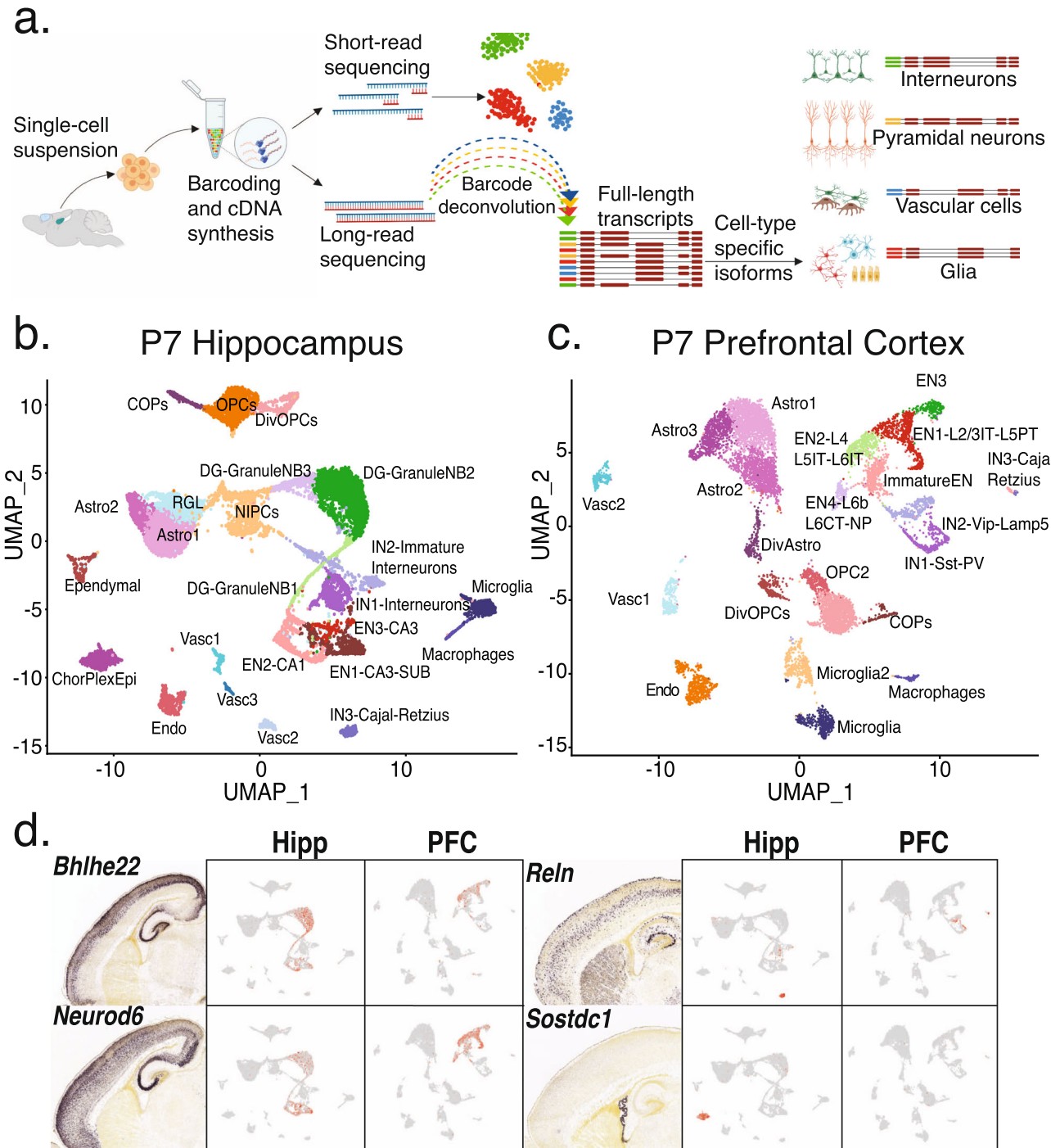

**Fig. 1 Short read clustering of P7 hippocampus and prefrontal cortex tissue recovers precursors to known adult cell-types. a** Schematic of the ScISOrSeq workflow (created with BioRender.com). **b** UMAP of P7 hippocampus (HIPP) data. Cell-types identified by marker genes, RNA velocity analysis, and alignment to published data shown in S1. **c** UMAP of P7 prefrontal cortex (PFC) data. Cell-types identified by marker genes, RNA velocity analysis, and alignment to published data shown in S2. **d** In situ hybridization (ISH) images from Allen developing mouse brain atlas for marker genes and corresponding projections on UMAP plots from HIPP and PFC.

abundance per gene across brain regions, which follow the same statistical framework as the isoform tests. 141 (of 395) genes exhibited differential TSS or polyA-site usage (Fig. 2g). By extension, the remaining 254 genes are explained by splice-site usage differences.

Many genes with DIE have $\Delta\Pi \leq 0.5$; however, we also identified drastic switches ($\Delta\Pi \geq 0.5$): *Nsfl1c* encodes the Nsfl1 cofactor p47, which regulates tubular ER formation, influences neuronal dendritic spine formation, and dendritic

arborization[37,38]. A 6nt microexon is preferentially included in HIPP across neuronal and glial cell types but is absent in the same PFC cell types (Fig. 2h). The synaptic gene *Nsmf* is involved in the cAMP pathway[39,40] and through nuclear translocation of its protein, in memory formation[39,40]. In our data, the major HIPP isoform is absent in PFC, while the second HIPP isoform represents the majority of that gene's PFC expression. The isoforms differ by a 69nt exon with a nuclear localization signal and one of two synaptic targeting elements.

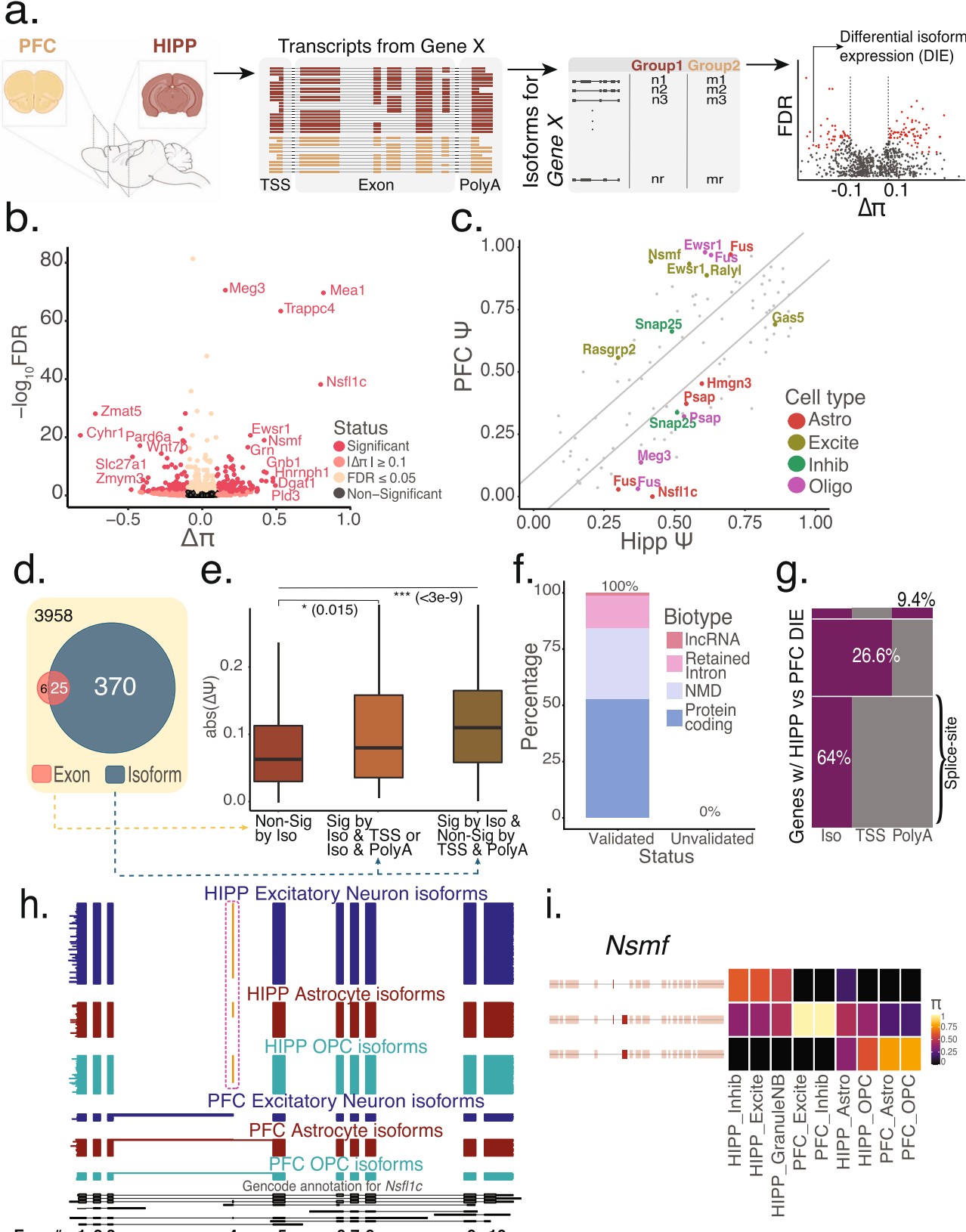

Hence, this exon may affect the synapse to nucleus signaling that the protein is involved in. A third *Nsmf* isoform with the 69nt exon but lacking a 6nt micro-exon is favored in PFC over HIPP but completely missing from neuronal cells, highlighting the regulatory role of micro-exons in neuronal function[1,41] (Fig. 2i).

**Differential isoform expression across brain regions is governed predominantly by one specific cell type.** Gene expression transcript similarities among clusters defined a cell-type hierarchy, first separated by neurons and non-neurons, and then by other cell types (Fig. S6). Since inhibitory neuron types are transcriptionally more similar to each other than to excitory

**Fig. 2 A gene-wise test for differential isoform expression (DIE) is more sensitive than an exon-wise test at detecting splicing changes. a** Schematic of the scisorseqR approach—Barcode deconvolution, filtering, pairwise comparison, and reporting of significant results based on FDR and $\Delta\Pi$ cutoffs (created with BioRender.com). **b** Volcano plot of bulk HIPP vs. bulk PFC differential abundance analysis, with the effect size ($\Delta\Pi$) on the $X$-axis. $P$-values derived from a $\chi^2$ test and corrected for multiple testing using the Benjamini–Hochberg correction are plotted on the $Y$-axis. Points are colored according to the levels of significance based on FDR and $\Delta\Pi$ value. Genes considered significant (pink) when FDR $\leq 0.05$ and $|\Delta\Pi| \geq 0.1$. **c** Scatter plot showing the $\Delta\Psi$ of all exons for genes that show significant DIE between HIPP and PFC. Gray points represent non-significant exons. Points are colored according to the cell-type in which an exon is considered significant by a BY corrected $p$-value and a $\Delta\Pi \geq 0.1$. Diagonal lines indicate cutoff of 0.1 $\Delta\Psi$. **d** Venn diagram showing the overlap of genes significant by DIE (BH correction) with genes significant by exon tests (BY correction). **e** Boxplot showing the maximum absolute value of $\Delta\Psi$ per gene in three different categories: genes that are not significant by DIE tests ($n = 1395$), genes that are significant by DIE tests and also exhibit differential TSS or polyA-site usage ($n = 38$), and genes that are significant by DIE and do not exhibit differential TSS or polyA-site usage ($n = 128$). $P$-values were calculated using a two-sided Wilcoxon rank sum test. Data are represented as boxplots where the middle line is the median, the lower and upper hinges correspond to the first and third quartiles, the upper whisker extends from the hinge to the largest value no further than 1.5× IQR from the hinge (where IQR is the inter-quartile range) and the lower whisker extends from the hinge to the smallest value at most 1.5× IQR of the hinge. Please see function geom_boxplot in R (ggplot2). **f** Percentage of novel transcripts by scisorseqR that were manually validated as being novel by Gencode team, and breakdown of predicted function. **g** Heatmap of significant DIE genes ($n = 395$, BH correction) according to entire isoform between bulk HIPP and bulk PFC that also exhibit differential usage of transcription start site (TSS) and polyA-site (PolyA). Each row is a single gene. Gray represents genes that are non-significant by category (Iso/TSS/PolyA) whereas purple represents significant by category. **h** Isoform expression of Nsfl1c gene. Each horizontal line in the plot represents a single transcript colored according to the cell-type it is represented in. Therefore, blocks represent exons and whitespace represents intronic space (not drawn to scale). Orange exon represents alternative inclusion. **i** Isoform expression for the Nsmf gene. Each row represents an isoform colored by $\Pi$ and each column represents a cell-type in HIPP or PFC.

types[34,35], we grouped finer inhibitory neuron subtypes (IN1, IN2, IN3) into a composite inhibitory neuron (IN) category, and excitatory neuron subtypes into an excitatory neuron (EN) category. We hypothesized three alternative models that could underlie differential isoform expression between brain regions: 1) multiple or all cell types change splice variants ('Both-Cell-Types-Model'); 2) a single cell type changes splice variants ('Single-Cell-Type-Model'); or 3) changes in expression or cell-type abundance without any change in splice variants ('No-Cell-Type-Model') (Fig. 3a).

We analyzed neuronal and non-neuronal reads separately and cross-referenced the results of the 395 genes with DIE. 26 genes (6.6%, FDR $<= 0.05$) had DIE in neurons and non-neurons under the Both-Cell-Types-Model, 151 (38.2%) in neurons only and 81 (20.5%) in non-neurons only under a Single-Cell-Type-Model, and 137 (34.7%) that were either too low in expression for testing (Methods) in both neuronal and non-neuronal cells or followed the No-Cell-Type-Model (Fig. 3b). To distinguish these two explanations for the 137 genes, we considered genes with $\Delta\Pi \geq 0.1$ irrespective of $p$-values for the 395 genes. Specifically, we calculated the ratio of $\Delta\Pi$ in a finer subtype to the $\Delta\Pi$ in the composite cell type. After dividing all cells (composite) into neurons and non-neurons (finer level), 75% ($\pm 2.3$, $SE_p$) of bulk DIE events were traced to only neurons or only non-neurons (Single-Cell-Type-Model, green) and 24.4% to both (Both-Cell-Types-Model, light purple, Fig. 3ci). A single gene followed the No-Cell-Type-Model (0.3%, dark purple). Similarly, when dividing neurons into excitatory and inhibitory subtypes, we found 78.8% ($\pm 2.97$) for the Single-Cell-Type-Model, 19.58% for Both-Cell-Types-Model, and 1% for No-Cell-Type-Model (Fig. 3cii). When separating composite non-neuronal cells into glia and vascular + immune cells, and again when separating glia into astrocytes and oligodendrocytes, and then the vascular + immune cluster into vascular and immune cells, we observe similar trends. The Single-Cell-Type-Model was more prevalent than the Both-Cell-Types and No-Cell-Type Models (Fig. 3c). These observations were replicated in replicate 2, despite differences in cell-type proportions between replicates (Fig. S7a–c). In summary, the No-Cell-Type-Model is rare, representing 0.3–3.28% for the different cell group divisions. Extending this observation to the 137 genes above (Fig. 3b), one gene (0.3% of 395) likely represents the No-Cell-Type-Model whereas the other 136 can be attributed to low read depth.

An example of the dominant Single-Cell-Type-Model is Hexosaminidase A (*Hexa*), which is implicated in Tay-Sach's disease[42]. In addition to a single annotated isoform, PFC excitatory neurons show significantly diminished inclusion of an internal exon (from 81% inclusion in HIPP to 22%), thus expressing a novel isoform. Other cell types show no difference between HIPP and PFC. Manual validation classified this novel isoform as NMD, indicating brain-region and cell-type-specific NMD (Fig. 3d).

**Brain regions can override cell-type specificity for a subset of genes, possibly through microenvironmental influence.** Despite the prevalence of the Single-Cell-Type-Model, the Both-Cell-Types-Model is still common. To avoid circular reasoning, we considered all genes sufficiently expressed (Methods) in neurons and non-neurons in both brain regions. Concurrent regional DIE differences ($\Delta\Pi \geq 0.1$) in both neurons and non-neurons occur more often than expected by chance ($\frac{\text{observed}}{\text{expected}} = \frac{23.9\%}{15.6\%} = 1.5, p \leq 2.2e^{-16}$, Fisher's two-sided exact test). This trend is also conserved in excitatory and inhibitory neurons ($p \leq 2.2e^{-16}$) as well as in glia and vascular + immune cells (Fig. 3e). Not only do we see concurrence in Rep1, but we find that the observation is conserved in all investigated levels across both replicates considered together (Fig. S7d). Two non-mutually exclusive models may underlie this observation—micro-environment and cell origin. Firstly, HIPP and PFC interneurons originate in the ganglionic eminences[31,33], while excitatory neurons do not. Thus, splicing similarities between HIPP EN and IN that are different from PFC EN and IN might be imposed by the regional microenvironment. Secondly, considering neurons and glia, their common descendance from radial glial stem cells may underlie cases of brain-region-specific regulation that overrides cell-type specificity.

**Cell types endogenous to one brain region have distinct splicing signatures.** We traced the contribution of individual cell types to bulk DIE. Region-specific DIE was clearly traceable ($\frac{\Delta\pi_{\text{cell-type}}}{\Delta\pi_{\text{Bulk}}} \geq 0.9$) in 73.4% ($n = 395$) of the cases, while 10.4% had $\frac{\Delta\pi_{\text{cell-type}}}{\Delta\pi_{\text{Bulk}}} \leq 0.9$ in all cell types (Fig. S7e, f). The remaining 16.2% had low read counts in all cell types. Some genes truly have a regional component i.e., concurrent changes across multiple cell types (Fig. S7f). The low depth and $\Delta\Pi$ ratio could arise from cell

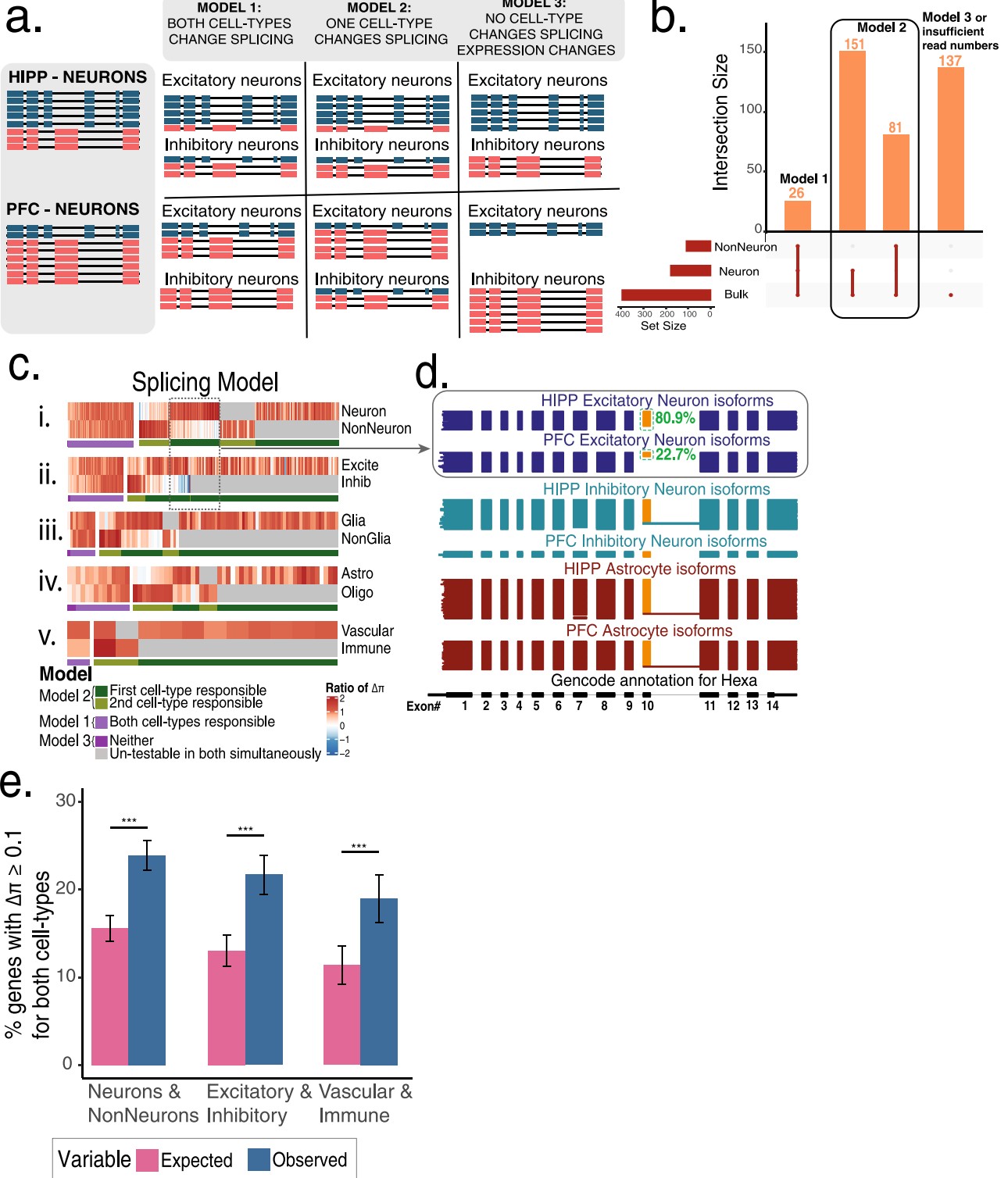

**Fig. 3 Three models of alternative splicing followed by differentially spliced genes across brain regions. a** Schematic of three models that explain splicing changes between any two categories (e.g., HIPP neurons versus PFC neurons). **b** Upset plot of DIE genes in bulk, neuron, and non-neuron. **c** Five gene × celltype heatmaps clustered by the ratio of ΔΠ of an individual cell-subtype to a parent cell-type i.e., value for Neuron = $\Delta\Pi_{Neuron}/\Delta\Pi_{Bulk}$. Each vertical line indicates the ratio of ΔΠ for a single gene. Gray lines indicate lack of sufficient depth or lack of expression. Clusters of genes are colored by whether both cell-types show similar relative ΔΠ to the parent (purple, Model I, Model III) or whether one cell-type explains most of the splicing changes (Model II). **d** Hexa gene representing an example of Model II. Each horizontal line in the plot represents a single transcript colored according to the cell-type it is represented in. Therefore, blocks represent exons and whitespace represents intronic space (not drawn to scale). Orange exon represents alternative inclusion. **e** Barplots indicating percent of genes with |ΔΠ| ≥ 0.1 for two concurrently assessed cell-types. Pink bar indicates expected levels assuming independence, while blue bars represent observed levels (n = 2351,1320,785). Error bars indicate 95% confidence interval of proportion. P-values calculated using Fisher's two-sided exact test.

types only present in HIPP but not PFC. Indeed, reads originating from granule NB, RGL, ependymal, and CPE clusters have a higher ratio in the 26.4% of genes for which DIE was not explained (gray) than in the genes where DIE was explained (yellow) (Fig. S7g). For example, for the *Fxyd1* gene, CPE cells in HIPP had a different splicing signature from astrocytes in HIPP and PFC, leading to regional DIE (Fig. S7h). These observations warrant further exploration of each cell type's splicing signatures.

**Choroid plexus epithelial cells (CPEs) generate distinct isoforms predominantly through alternative TSS.** We performed DIE tests in pairwise comparisons of HIPP cell types. DIE was most frequent for neuron vs. non-neuron comparisons in HIPP (Fig. 4a), and this was confirmed in PFC (Fig. S8). High percentages were also seen in some comparisons between non-neuronal cell types. Interestingly, comparisons between non-neuronal cell types showed higher DIE than those observed within neurons (Fig. 4b). Importantly, non-neuronal comparisons involving CPEs clearly had the highest DIE fractions. CPEs are cerebrospinal fluid secreting ependymal cells in cerebral ventricles, and alternative splicing in CPEs relates to disease[9,43] (Fig. 4c). Surprisingly, TSS choice (compared to exons and polyA-sites) largely explained the isoform regulation of CPE cells (Fig. 4d). Furthermore, CPE-associated transcripts strongly favored an upstream TSS compared to the non-CPE transcripts (70/93 genes, Bernoulli $p = 3 \times 10^{-7}$). This can allow for CPE-specific post-transcriptional modifications[44], translation initiation, and transcription factor control of gene expression (Fig. 4e).

**Single-cell basis of DIE between cell types.** When DIE is observed between two cell types, two competing hypotheses can explain this phenomenon[4]. Either all cells of each cell-type behave uniformly and reflect the differences in isoform expression between the two cell types, or individual cells of one or both cell types could show variability in isoform expression. Neuronatin (*Nnat*) is an important developmental gene expressing a neuron-specific isoform. In *Nnat*, DIE between ependymal cells and excitatory neurons is represented by the vast majority of individual cells. However, the case of DIE between excitatory neurons and granule neuroblasts is different: some granule neuroblasts behave like excitatory neurons, while others behave like non-neurons. This may be due to different sub-populations of granule neuroblasts (Fig. 4f).

**Clustering on long-read data recapitulates short-read cell-type assignments.** We clustered hippocampal cells using their isoform expression similarities. Compared to 3′seq clustering, glial, immune, and vascular clusters were similarly defined (Figs. S9a, b and 4a). Jaccard similarity index analysis between short-read and long-read clusters showed high concordance for broad-level classification (Fig. S9c). Additionally, isoforms of some genes are better resolved with long reads than with than 3′-expression short reads, including for distinguishing neurons and non-neurons (e.g., *Pkm, Clta, H3f3b*), or mature and immature neurons (e.g., *Cdc42, Srsf3, Thra*, Fig. S9d). However, despite striking differences in isoform expression within neuronal subtypes, differences between isoform-derived clusters and short-read derived clusters remained. Long-read clustering successfully separated CA1 from CA3 neurons (i.e., short-read EN1 vs EN2) but did not separate all cells of the more immature IN2 cluster from mature granule neuroblasts i.e., GranuleNB-2 (Fig. S9e, f). Such differences could be explained by cell subtype specificity in isoforms only, or reduced sequencing depth of isoforms.

**Relative isoform expression differences during development reflect dynamic changes in function.** Using Slingshot[45] on a subset of hippocampal cells, we recovered the radial-glia-like (RGL) to excitatory-neuron developmental lineage. From RGLs to NIPCs only 5.1% of tested genes showed DIE ($n = 73$; 95% CI = [4.03,6.34]). However, threefold more did so from NIPCs to GranuleNB ($n = 359$, 95% CI = [15.81,19.10]) and then from GranuleNB to excitatory neurons ($n = 423$, 95% CI = [14.72,17.54]) (Fig. 5a). Gene ontology (GO) analysis[46,47] revealed isoform changes in the splicing machinery itself in earlier steps, i.e., from RGL to NIPCs (for *Hnrnpa2b1, Snrnp70, Srsf2, Srsf5, Srsf6, Srsf7, Rbm3*; Fig. 5b). However, as granule neuroblasts matured to excitatory neurons, DIE was associated with synapse formation and axon elongation (*Snap25, Snca, Syp, Dbn1, Cdc42, Nptn, Gap43*) among others (Fig. 5c, d).

Importantly, many exon inclusion levels are altered in the transition from dentate gyrus (DG) granuleNB to more mature EN in CA1 and CA3. The synaptic Calcium/Calmodulin-dependent protein kinase II Beta (*Camk2b*) has enzymatic and structural roles in neuronal plasticity[48]. Embryonic *Camk2b* exploits a 3nt addition to exon 11 and exon-12 exclusion, translating an Alanine instead of Valine[49]. The embryonic form (red exon-11 extension, exon-12 exclusion, Fig. 5e) dominates in more immature granuleNB2 compared to granuleNB1. However, this isoform persists infrequently in mature neuronal types (EN1, EN2, and IN1) indicating cell-type specificity during developmental regulation. Moreover, the additional 3 nucleotides (red in exon 11) co-occur with exon 12, which has not been reported. Also, the first alternative exon (exon 9) increases inclusion as cells differentiate. Furthermore, exon 9 coordination with exon 12 defines cell-subtype differences between EN and IN. All three splicing events occur in the actin binding domain of the CaMKIIβ structure and encode several confirmed phosphorylation sites[50]. Thus, exon coordination among distinct cell types could indicate cell-type-specific morphological changes in the actin cytoskeleton, for instance in spine dynamics[51] (Fig. 5e).

For the post-synaptic *Dlgap4* linked to neuropsychiatric disorders[52] non-neurons and immature neurons predominantly express one exon. However, during neuronal maturation, exon inclusion switches to both exons (Fig. S10a, b). For *Nptn*, a gene involved in long-term potentiation, glia and immature neurons predominantly express a single 9nt micro-exon. Mature neurons employ an upstream acceptor, adding four amino acids encoding the cytoplasmic domain, likely relevant for protein–protein interactions[53,54] (Fig S10b).

**Hippocampus enriched developmental gene Fibroblast growth factor 13 (*Fgf13*) shows neuronal subtype-specific TSS.** *Fgf13* exhibited high DIE ($\Delta\Pi > 0.5$) across multiple neuronal cell-type and subtype comparisons. *Fgf13* reaches peak expression, specifically in HIPP, at our investigated time point (P7)[55]. *Fgf13* is a neuronal developmentally regulated gene and lethal when knocked-out[56,57]. A member of the fibroblast growth factor (FGF) superfamily, *Fgf13* is one of four FGF family members (*Fgf11-14*) labeled fibroblast growth factor homologous factors, which—unlike most FGFs—do not have signal sequences, are not secreted, and function intracellularly[58]. Among its intracellular roles are regulation of voltage-gated sodium channels[59–61], rRNA transcription[62], and microtubule stabilization[55]. Of the various *Fgf13* isoforms[63], two isoforms with distinct TSS dominate during brain development[55,64]. We find that *Fgf13* is particularly alternatively spliced between excitatory and inhibitory neurons (Fig. 5f). The downstream-TSS isoform, Fgf13-S, is the major HIPP isoform across all excitatory types, and immature inhibitory

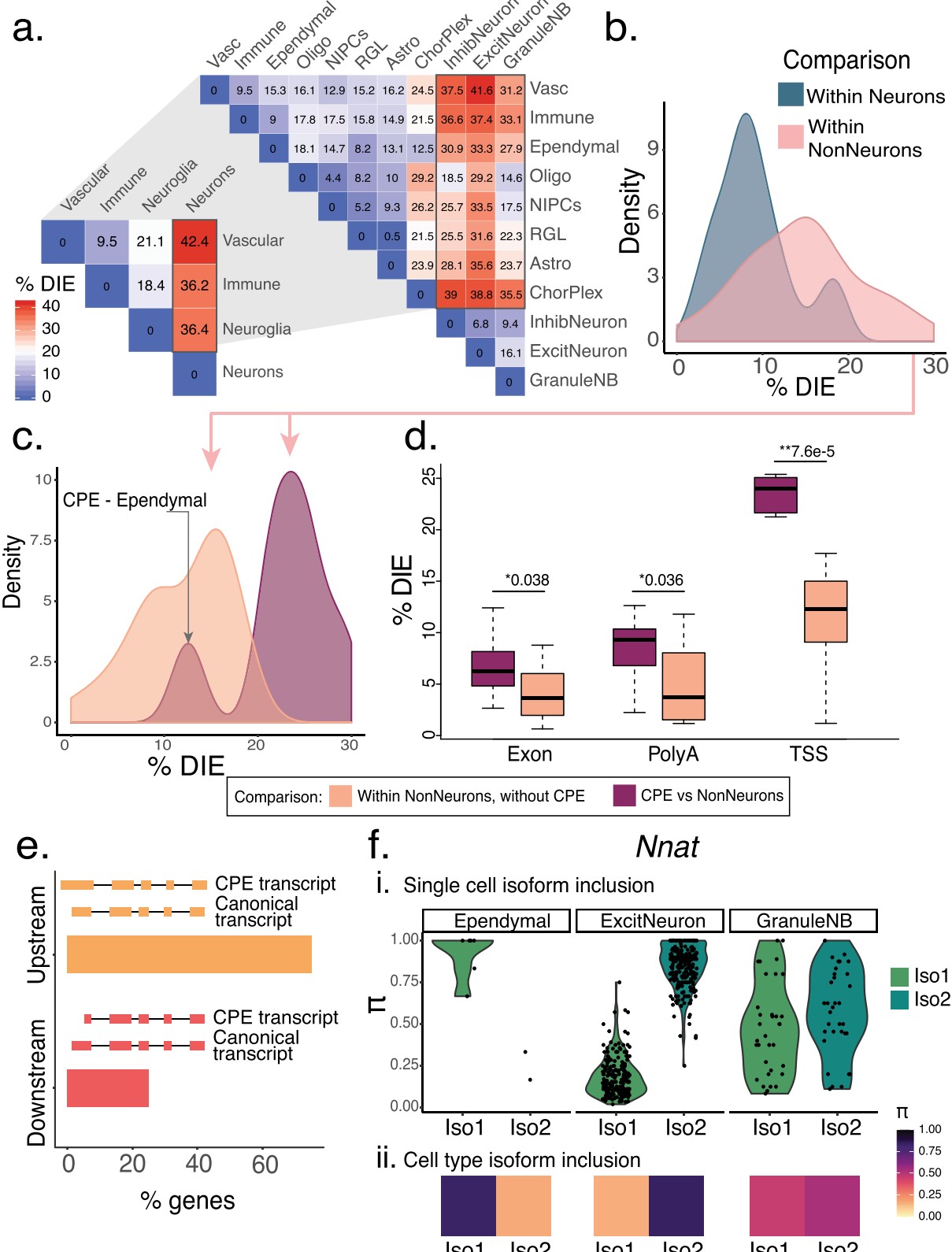

types. Conversely, the upstream-TSS isoform, Fgf13-VY, is partially seen in DG neuroblasts and dominates in inhibitory interneuron classes. This was confirmed using Basescope analysis with probes designed for excitatory and inhibitory neuron marker genes, and separate probes for the S and VY isoforms. We find that the S isoform (pink) co-localizes with excitatory neurons (green, *Neurod6*) and not inhibitory neurons (green, *Gad2*),

whereas the VY isoform (pink) co-localizes with the inhibitory neurons but not the excitatory neurons (Fig. 5g, Methods). The isoforms also differ in subcellular localization: reflecting its role in regulating ribosomal biogenesis[62], Fgf13-S is primarily localized to the nucleolus, whereas Fgf13-VY is present throughout the cytoplasm, consistent with its known role in regulating voltage-gated sodium channels[60] (Fig. 5h).

**Fig. 4 Choroid plexus epithelial cells (CPEs) generate distinct isoforms predominantly through alternative TSS. a** Triangular heatmap representing percentage of significant DIE in pairwise comparisons at two levels of granularity. At the broad level: pairwise comparisons of neurons, non-neurons, immune cells, and vascular cells. Zooming in, at the narrow level: neuronal and non-neuronal categories are broken up into their constitutive cell-subtypes. **b** Density plot of percentage of significant DIE in pairwise comparisons from 4A, broken down by two categories: within neurons and within non-neurons. **c** Density plot showing DIE within non-neurons (pink region, Fig. 4b) broken up further by comparisons that either include (purple) or exclude (orange) choroid plexus epithelial (CPE) cells. **d** Boxplots of percentage significant genes in non-neuronal comparisons including and excluding the CPE, broken down by part of the transcript (TSS/splice-site/PolyA) responsible for splicing changes. X-axis indicates the part of the transcript assessed. P-values were calculated using a two-sided Wilcoxon rank sum test (***$p < 10^{-15}$, **$p < 0.001$, *$p < 0.05$). Data are represented as boxplots where the middle line is the median, the lower and upper hinges correspond to the first and third quartiles, the upper whisker extends from the hinge to the largest value no further than 1.5 × IQR from the hinge (where IQR is the inter-quartile range) and the lower whisker extends from the hinge to the smallest value at most 1.5 × IQR of the hinge. Please see function geom_boxplot in R (ggplot2). **e** Barplot showing the percentage of genes (n-90) for which the CPE transcripts are either upstream or downstream of non-CPE transcripts. **f** Percent inclusion (Π) of two *Nnat* isoforms across three cell-types: ependymal cells, excitatory neurons, and granule neuroblasts i. Violin plots of Π values of each isoform per single cell ii. Π values per isoform across all single cells.

---

**Synaptic genes associated with vesicle transport show splicing differences in developing hippocampal neurons.** Many synaptic genes have low expression in developing HIPP. However, sufficiently expressed (>50 reads) genes' isoforms distinguish neuronal subtypes. Two mutually exclusive exons define two isoforms (Snap25-a, Snap25-b) for Snap25, involved in endocytosis. Concordant with the literature[65–67], our CA3 excitatory neurons (EN3) have a higher proportion of Snap25-b transcripts than in CA1 and subiculum (EN1, EN2). However, our GranuleNBs have higher abundance of Snap25-a whereas mature excitatory neurons switch to Snap25-b. Interestingly, interneuron precursors and Cajal–Retzius cells (IN1,IN3) rely more on Snap25b than their excitatory counterparts, and thereby seem to rely more heavily on larger primed vesicle pools[68] (Fig. S11b). Alternatively, interneurons could switch from Snap25-a to Snap25-b before excitatory neuron development.

We also observe synaptic genes with key splicing differences between neurons and non-neurons. Clathrin light chain A and B (*Clta*, *Cltb*) work alongside *Epsin1* in vesicle-mediated endocytosis[69]. For all three genes, exon inclusion distinguishes neurons from non-neurons and granuleNB from mature EN. For *Clta*, an additional exon distinguishes neuronal subtypes (Fig S12a, b). The neuronal specific insertions in the clathrin light chain may influence the association with the slow axonal transport of clathrin[70].

**Slide isoform sequencing (sliso-Seq) to delineate spatial localization of splicing changes.** To ground our observations in a spatial sense, we generated 10X Genomics Visium spatial transcriptomics data from a P8 sagittal section. Alignment with single-cell short-read HIPP data confirmed the spatial localization of excitatory neurons in the CA regions and subiculum (Fig. 6a), and alignment with the PFC data confirmed the excitatory precursors in distinct cortical layers (Fig. 6b). We then devised a long-read sequencing approach for spatial transcriptomics (Methods). To validate regional specificity of isoform expression using orthogonal techniques, we correlated ΔΨ values between composite HIPP and composite PFC single-cell data with ΔΨ values of hippocampus vs. cortex using long-read sequencing. Based on single-cell data, we focused on 40 exons with region-specific splicing patterns and without alternative acceptors/donors (Methods). Overall, we found strong concordance: for 85%, both single-cell and spatial HIPP vs. PFC splicing differences point in the same direction (Bernoulli probability < = 3.5e-06) (Fig. 6c). Additionally, we confirmed neurodevelopmental exon inclusion switches in *Pkm* and *Clta*, where the non-neuronal and developmental exons from the single-cell data were enriched in the DG and in the choroid plexus of the spatial data (Fig. S12, Fig. 6d). For *Snap25*, the neurodevelopmental switch from Snap25-a to Snap25-b in single-cell data (compare Fig. S11b),

occurs in a posterior-to-anterior gradient in spatially mapped exons (Fig. 6e). This supports the idea that the microenvironment can dictate brain-region-specific splicing for some genes. Also, the hypothesis that interneurons selectively switch isoforms before excitatory neurons seems unlikely.

## Discussion

Temporal and anatomic differences in alternative splicing are implicated in developmental changes in molecular function[71]. Building on the short-read investigation of cell-type-specific alternative splicing in the brain[72], our data enable the illumination of full-length isoform regulation across cell types. We endeavored to generate cell-type, brain-region, and age-specific maps of AS to understand functional consequences of differential isoform expression (DIE). Our analysis hinges on a single-per-gene isoform test that considers a gene's entire isoform repertoire and outperforms exon-based tests. Region-specific isoform testing revealed 395 genes with DIE between hippocampus and PFC, partially caused by isoforms novel to GENCODE. Manual validation by the GENCODE team using rigorous metrics lends credibility to this novel isoform detection. Thus, filtering these isoforms using bulk short reads, CAGE and polyA-site data provides a mechanism to automate genome annotation for isoforms.

We defined cell types underlying DIE between brain regions. In most cases and levels of granularity, we identified a finer cell type explanation of DIE, including the altered part of the transcripts. Despite inhibitory interneurons in HIPP and PFC migrating in from a common origin[31], we observe a signature of coordinated DIE in excitatory and inhibitory neurons for a gene subset. Such gene subsets with coordination across cell types were observed at all investigated levels. Microexons are linked to neuronal cell types for their regulatory function[1,41], however, in the case of *Nsfl1c* we find that their expression is not limited to neurons but instead exhibits regional regulation by being expressed only in HIPP. Thus, brain region can override cell-type specificity for a subset of genes, which may explain region-specific sQTLs[21]. The theoretical possibility that brain-region DIE in bulk arises purely from cell-type abundance or transcriptional activity differences is rarely observed. However, to which extent these observations persist in case–control studies of disease, across adult brain regions, and across species requires further studies.

Our results indicate that understanding the cell-type basis of brain-region-specific DIE requires a thorough understanding of cell-type-specific DIE within each brain region. This further warrants a brain-wide map of isoform expression at a single-cell level. Within the brain, isoform diversity in non-neuronal cell types has attracted less attention than in their neuronal counterparts. However, we find that non-neuronal cell types exhibit high pairwise DIE. This may be partially due to the functional

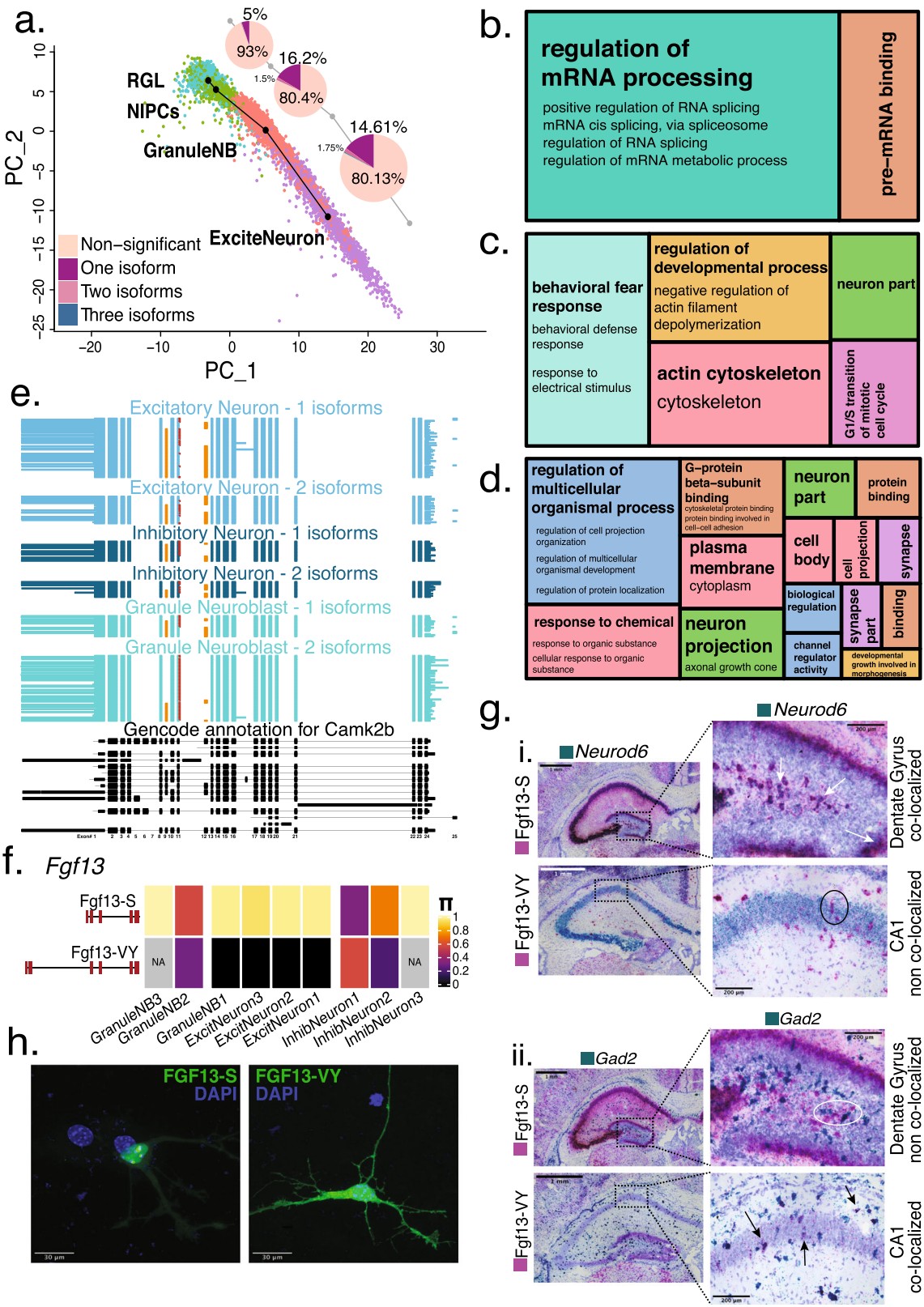

diversity of non-neuronal cells, or due to origins from different stem cell populations. Importantly, excitatory neurons and astrocytes which originate from the same stem cell population but are functionally distinct show high DIE (35.8%). However, the excitatory neuron population also shows high DIE with Cajal–Retzius cells (18%) despite both originating from different stem cell populations but functioning as neurons. Among the non-neurons, choroid plexus epithelial cells show particularly large differences from other non-neuronal (but also neuronal) cell types. Surprisingly this observation is largely caused by CPE-specific choices of predominantly upstream TSSs. This raises the question whether other highly specialized cell types in other brain areas exhibit similar complex alternative transcriptome mechanisms.

**Fig. 5 Relative isoform expression differences during development reflect dynamic changes in function. a** Splicing changes seen at every transition step of neuronal differentiation trajectory. Pie chart indicates the number of isoforms needed to be considered in order to reach the $|\Delta\Pi| \geq 0.1$ threshold for a gene to be considered significantly DIE. **b** Treemap of condensed gene ontology (GO) terms for the transition from RGL to NIPCs in the neuronal differentiation trajectory, with size of boxes corresponding to number of significant terms associated with the GO category. **c** Treemap of condensed GO terms for the transition from NIPCs to GranuleNB (GNB1, GNB2, GNB3). **d** Treemap of condensed GO terms for the transition from NIPCs to Excitatory Neurons (EN1, EN2, EN3). **e** Hippocampal cell-type specific isoform expression in *Camk2b* gene. Each horizontal line in the plot represents a single transcript colored according to the cell-type it is represented in. Orange exons represent alternative inclusion, red extension of exon 11 represents a 3nt addition. **f** Π value of the S and VY isoforms for *Fgf13* gene across hippocampal neuronal cell types. **g** Basescope (in situ hybridization) images ($N = 1$, repeated for 2 technical replicates). Data showing *Fgf13*-S and *Fgf13*-VY isoform expression (pink stain) in the hippocampus, with simultaneous staining for excitatory neurons (*Neurod6*—green) and inhibitory neurons (*Gad2* - green). Scale bar, 1 mm. Each plot features enlargements of the dentate gyrus for the S isoform, and CA1 region for the VY isoform with arrows indicating co-localization while circles indicate lack of co-localization. Scale bar, 200 μm. **h** Representative images of subcellular localization of overexpressed GFP-tagged Fgf13-S isoform in nucleolus and GFP-tagged Fgf13-VY isoform in cytoplasm. ($N = 3$, 20–30 GFP + neurons examined per transfection). Scale bar, 30 μm.

We notice the sheer functional diversity that AS lends to the splicing machinery, synaptic plasticity, and vesicle-mediated endocytosis. This motivates further investigation linking a spliceosomal-gene isoform to the splicing of its target gene, as well as the isoform state of synaptic genes in neuronal subtypes. The evidence for isoforms adding to cellular diversity is further bolstered by long-read clustering, which yields coherent cell-types albeit with discrepancies, which could be partially due to the sparsity of the isoform matrix due to lower long-read throughput. Alternatively, cell states could be defined by an isoform expression program that does not correspond to 3′seq-based cell-type definition, highlighting the need for factoring in isoform expression to re-define traditional transcriptional cell-types.

Finally, we have devised the method of slide-isoform sequencing (slISO-Seq), which employs spatial transcriptomics and long-read sequencing. This allows anchoring the above results in a spatial view of the brain and reveals important biology such as the brain-wide coordination of the *Snap25* isoform switch. Looking forward, the integration of long-read single-cell and spatial technologies allows for the possibility of constructing 3-dimensional maps of isoform expression at single-cell resolution.

In summary, these results present a detailed view of cell-type-specific full-length isoforms across brain regions, bringing us closer to a comprehensive isoform map of the brain. The software generated here is employable in a much larger setting and available as an R-package. It extends to case–control studies, broad sample comparisons, and spatially anchored comparisons beyond mouse or brain.

## Methods

**Ethics statement**. All experiments were conducted in accordance with relevant NIH guidelines and regulations, related to the Care and Use of Laboratory Animals tissue. Animal procedures were performed according to protocols approved by the Research Animal Resource Center at Weill Cornell College of Medicine.

**Animals and tissue isolation**. C57BL/6NTac ($n = 6$) female pups were quickly decapitated. For the single-cell experiments (P7; Rep1 $n = 1$, Rep2 $n = 2$), the brains were removed and placed on a stainless-steel brain matrix for mouse (coronal repeatable sections, 1 mm spacing), and the prefrontal cortex and hippocampus was dissected[73] in cold PBS solution (Fig. S13a–f). Brain tissues from both hemispheres were pooled into one sample. After dissection, tissues were snap frozen in dry ice until processing. For the 10X Visium spatial experiment (P8; $n = 1$) brains were fresh-frozen and embedded in OCT. For the Basescope (P7; $n = 2$) experiments, the brains were transcardially perfused, immersion fixated and cryo-protected (15% and 30% Sucrose in phosphate-buffered saline) each over night before being embedded in OCT. Mice were housed in an air-conditioned room at 65–75 °F (~18–23 °C) with 40–60% humidity and a 12-h-light, 12-h-dark cycle. All animal experiments were approved by the institutional animal care and use committee (IACUC) of Weill Cornell Medicine

**Tissue disassociation**. Following recommendations from 10x Genomics (Cat#CG00055 Rev C) dissected hippocampus and prefrontal cortex tissue was placed into 2.5 ml Hibernate E/B27/GlutaMax medium (BrainBits cat#HEB) at Room Temp until all samples were dissected. HEB medium was removed and replaced with 2 ml of 2 mg/ml activated papain (BrainBits cat#PAP) then incubated for 25 min at 37 °C with gentle mixing. After allowing tissue to settle, papain was removed and replaced with 2 mL fresh HEB medium and tissue was gently triturated 15–20 times using a wide-bore pipette tip and tissue left to settle. Supernatant was taken and filtered using a 30 μm cell strainer (Miltenyi Biotec cat#130-041-407) into a collection tube. To the remaining tissue, another 2 ml of fresh HEB medium was added and then triturated with a regular 1 ml pipette tip an additional 10–15 times until tissue was completely disassociated. Supernatant was taken and filtered through a 30 μm cell strainer and added to the collection tube. Supernatant was then centrifuged at 400rcf for 2 min. The cell pellet was re-suspended in 1–3 ml of neuronal culture medium NbActiv1 (BrainBits cat#Nbactiv1) depending on cell pellet size, filtered through a 30 μm cell strainer (Miltenyi Biotec cat#130-041-407), and was subsequently diluted to 1500 cells/μl in NbActiv1 for capture on the 10x Genomics Chromium controller.

**10x Genomics single-cell capture**. The disassociated cells were captured on the 10x Genomics Chromium controller according to the Chromium Single Cell 3′ Reagent Kits V2 User Guide (10x Genomics CG00052 Rev F) with the following modification. PCR cycles were increased, from the recommended ten cycles for recovery of 8000 cells, to 16 cycles to target a yield of cDNA enabling simultaneous Illumina and PacBio library preparation.

**Illumina and Pacific Biosystems library preparation**. Illumina library preparation was performed using 100 ng of amplified cDNA following the Chromium Single Cell 3′ Reagent Kits V2 User Guide (10x Genomics CG00052 Rev F) reducing final indexing PCR cycles to ten cycles from the recommended 14 cycles to increase library complexity. Sequencing for Replicate 1 was performed on HiSeq4000 according to 10x Genomics run mode, for Replicate 2, sequencing was performed on a NovaSeq S1 flowcell also following 10x Genomics run mode and Bulk RNA-Seq was performed on the Illumina NextSeq 500 with a 150 PE run mode. PacBio library preparation was performed with 500 ng of amplified cDNA using SMRTbell Express Template Prep Kit V2.0 (PacBio cat#PN: 100-938-900) to obtain Sequel II compatible library complex and was sequenced on a total 24 Sequel I SMRTcells with a run time of 10 h and 20 Sequel II SMRTcells with a run time of 30 h across samples and replicates.

**Modification of 10x Visium Illumina library preparation**. Illumina compatible libraries were made from Visium-derived cDNA using a modified protocol derived from NEB Ultra II DNA FS kit (NEB #E6177). Visium-derived spatial cDNA was diluted to 100 ng in 26 μl and combined with 7 μl NEBNext Ultra II FS Reaction Buffer and 2 μl NEBNext Ultra II FS Enzyme Mix and incubated at 37 °C for 15 min, 65 °C for 30 min to obtain fragmented, end-repaired, and A-Tailed cDNA. Samples were then subjected to double-sided size selection by following the Beckman Coulter SpriSelect (Cat# B23318) protocol with an initial ratio of 0.6x SpriBeads. Supernatant was then taken and additional SpriBeads added for a final ratio of 0.8x eluting to 35 μl EB Buffer. Adaptor ligation using 10x Genomics protocol was performed by combining 2.5 μl 10x Adaptor Mix with 30 μl NEBNext Ultra II Ligation Master Mix and 1 μl NEBNext Ligation Enhancer with the previously end-repaired cDNA and incubated at 20 °C for 15 min. Single-sided SpriBead cleanup was performed using a 0.8x ratio and eluted in 15 μl EB Buffer. Finally cDNA library was amplified by combining adaptor ligated cDNA with 5 μl 10x Genomics i7 Barcoded primer and 5 μl of 1:5 diluted 10x Genomics SI Primer and 25 μl NEBNext Ultra II Q5 Master Mix and cycled with the following thermocyler profile: 98 °C 30 s, 12 cycles of 98 °C 20 s, 54 °C 30 s, 72 °C 30 s, then final extension of 65 °C for 5 min, 10 °C hold. Amplified library was again subjected to a double-sided size selection using an initial ratio of 0.6x SpriBeads, supernatant was then taken and additional SpriBeads added for a final ratio of 0.8x eluting to 35 μl EB Buffer. Illumina Libraries were then checked for quality and sequenced on a Illumina NextSeq500 instrument according to guidelines.

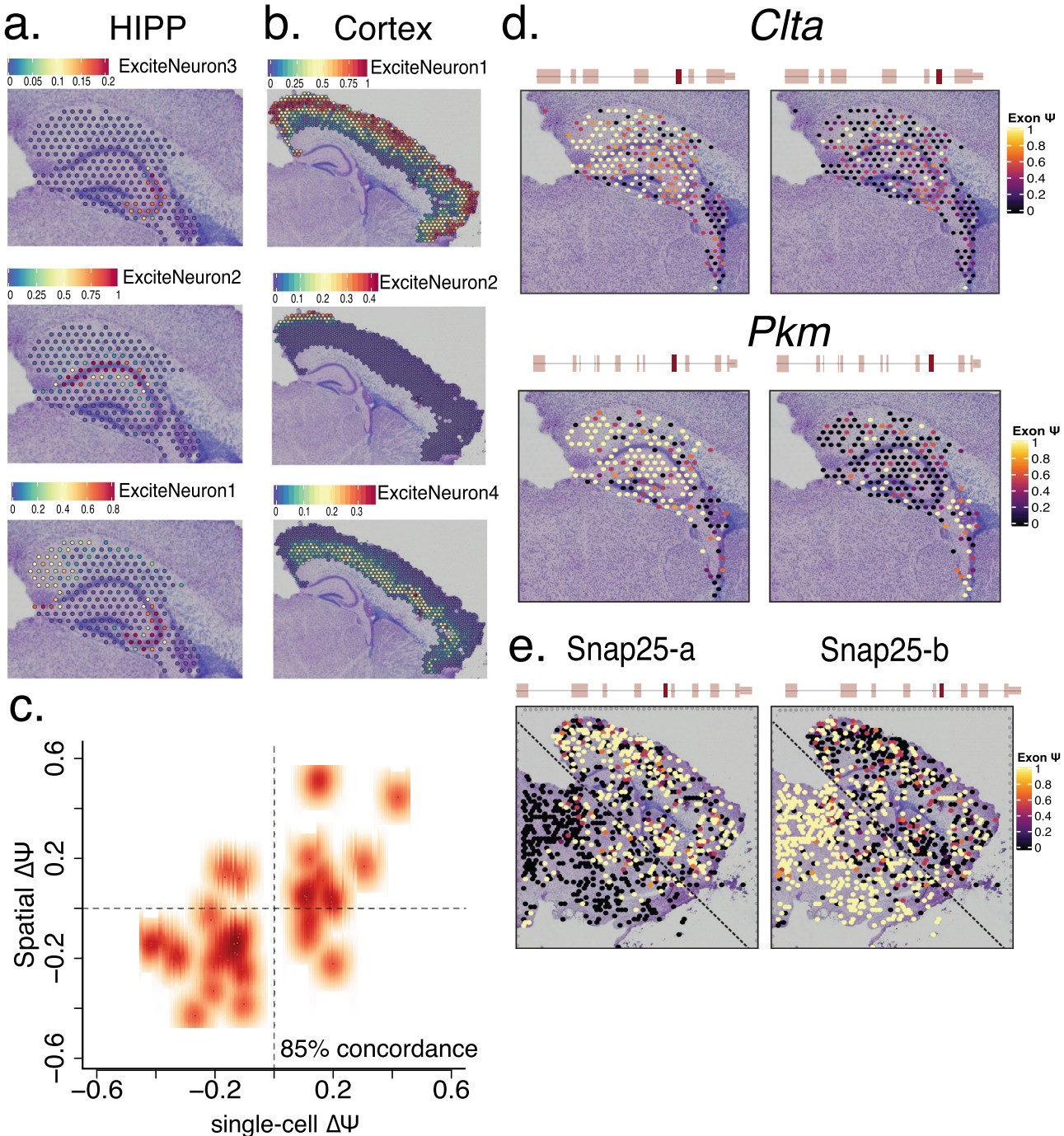

**Fig. 6 Slide isoform sequencing (sl-ISO-Seq) confirms spatial localization of splicing changes. a** Spatial localization of hippocampal single-cell excitatory neuron subtypes using gene-expression similarities to HIPP short reads. Color of spot indicates percentage of transcripts corresponding to indicated cell type. **b** Spatial localization of cortical excitatory neuron subtypes using gene-expression similarities to PFC short reads. Color of spot indicates percentage of transcripts corresponding to indicated cell type. **c** Scatter plot of the $\Delta\Psi$ between HIPP and PFC from the single-cell data, and the $\Delta\Psi$ between HIPP and cortex from 10X Visium spatial data ($r^2 = 0.6$). **d** Spatial distribution of alternatively spliced exons in *Clta* and *Pkm* genes in the hippocampal and choroid plexus regions. Color of spot indicates $\Psi$ values for each exon. **e** Spatial distribution of two mutually exclusive, alternatively spliced exons in *Snap25* gene across the whole slide. Color of spot indicates $\Psi$ values for each exon.

**Size selection of Visium cDNA using blue pippin for Oxford Nanopore**. Visium-derived cDNA was size-selected using Sage Sciences Blue Pipin instrument to obtain cDNA fragments 800 bp to 6000 bp for efficient sequencing on Oxford Nanopore PromethION Instrument. Samples were diluted to 1000 ng in 30 µl of TE buffer and combined with 10 µl of Sage Loading Solution before loading into one lane of a 0.75% Agarose Blue Pippin Cassette (Cat# BLF7510). Samples were then separated according to protocol for a target range of 800–6000 bp, and target elution retrieved after 12 h. Samples were then cleaned by Beckman Coulter

SpriSelect Beads (Cat# B23318) using a 0.8x ratio and eluted 50 µl Nuclease Free Water. Size-distribution was checked using Agilent Fragment Analyzer Large Fragment Kit (Cat# DNF-464-0500).

**PromethION library preparation and sequencing of Visium cDNA**. Oxford Nanopore compatible library was produced using 350 ng of either Sage Blue Pippin size-selected cDNA or non-selected cDNA derived from 10x Genomics Visium

following the Genomic DNA by Ligation protocol (SQK-LSK109) from Oxford Nanopore with the following modifications. End-repair was carried out omitting NEBNext FFPE DNA Repair and incubations extended to 10 min at 20 °C and 10 min at 65 °C. Loading inputs on the PromethION was increased to 150 fmol and sequenced for 20 h, and basecalling was done using Guppy (3.2.10).

**Total hippocampus and PFC short-read Illumina library preparation.** Illumina compatible libraries were produced from 1250 ng total RNA using NEBNext Ultra II RNA Library Prep Kit (NEB Cat#E7770S) following manufactures protocol with the following modifications. Target insert size was 450 bp for compatibility with paired end 150 bp sequencing mode. Number of PCR cycles was reduced to 6 to limit the effect of PCR aberrations on the final library. Sequencing was performed on the Illumina NextSeq 500 instrument.

**Generation of circular consensus reads.** Using the default SMRT-Link parameters, we performed circular consensus sequencing (CCS) with IsoSeq3 with the following modified parameters: maximum subread length 14,000 bp, minimum subread length 10 bp, and minimum number of passes 3.

**Primary hippocampal culture and transfection.** Primary dissociated hippocampal cultures were prepared as previously described[74], with minor modifications. Briefly, the hippocampus from P0 mouse pups was dissected on ice, digested with 0.25% trypsin for 30 min at 37 °C with DMEM (Sigma), and dissociated into single cells by gentle trituration. The cells were seeded at a density of 2.5–3.0 × 10^5 cells per coverslip in Neurobasal-A (Sigma) supplemented with 10% (vol/vol) heat-inactivated FBS onto coverslips previously coated with 50 µg/mL poly-D-lysine (Sigma) overnight at 4 °C and 25 µg/mL laminin (Sigma) for 2 h at 37 °C. The cells were maintained in a humidified incubator in 5% CO$_2$ at 37 °C. After 24 h, the medium was replaced with Neurobasal-A supplemented with 2% B27 (Invitrogen), 1% FBS, 25 µM uridine, and 70 µM 5-fluorodeoxyuridine. After 6 days of in vitro (DIV) culture, the neurons were transiently transfected with 0.2 µg of plasmid DNA per coverslip using calcium phosphate. One day after transfection cultured hippocampal cells were fixed for 30 min with 4% paraformaldehyde, washed three times with PBS, and incubated for 5 min in DAPI solution. Imaging was performed with a Zeiss LSM 880 Laser Scanning Confocal Microscope using an oil immersion 63× objective. All images were collected at a 2,048 × 2,048-pixel resolution. EGFP fusion constructs were generated as previously described[74].

**Alignment of bulk short-read data for junction validation.** Illumina short reads for HIPP and PFC were aligned to the reference genome (mm10) using STAR using the following parameters:
-outFilterMultimapNmax 1 -outFilterIntronMotifs RemoveNoncanonical
-outFilterMismatchNmax 5 -alignSJDBoverhangMin 6 -alignSJoverhangMin 6
-outFilterType BySJout -alignIntronMin 25 -alignIntronMax 1000000
-outSAMstrandField intronMotif -outSAMunmapped Within -runThreadN 32
-outStd SAM -alignMatesGapMax 1000000

**Alignment of single-cell short read data and analysis.** The 10x cellranger pipeline (version 3.0.0) was run on the raw Illumina sequencing data to obtain single-cell expression matrices. For replicate 1, the raw expression matrices obtained through cellranger were used along with the DropletUtils package (v 1.6.1)[75] to acquire 'eligible' barcoded single cells (FDR < = 0.001) with UMI counts that fell below cellranger's filtering cutoff. These barcodes were incorporated into new matrices for importing into Seurat (v 3.1). For both hippocampal replicates and the first PFC replicate, cells that had unique gene counts over 5000 or less than 700, and greater than 20% mitochondrial gene expression were removed from further analysis. To adjust for the lower mean reads/cell for the second PFC replicate, the cutoff for minimum number of genes per cell was lowered to 350. Filtering on these parameters yielded 14,433 single cells for the hippocampus across two replicates, and 10,944 single cells for the PFC. We then used Seurat's "merge" feature[76] to combine the replicates for each brain region. The number of UMIs, percentage of mitochondrial gene expression were regressed from each cell and then the gene expression matrix was log normalized and scaled to 10,000 reads per cell. Next, we clustered all the cells using 30 principal components (PCs) using the Louvain algorithm with a 0.6 resolution.

**Alignment of spatial short read data and analysis.** The 10X spaceranger pipeline was run on raw Illumina sequencing data to obtain spatial expression matrices. Seurat's spatial analysis functions were used to obtain gene expression similarity clusters and identify barcodes corresponding to various brain regions.

**Integrated analysis with published data to identify cell-types.** Published RNASeq P30 mouse brain data from Allen Brain Atlas[30] was used as a reference to identify cell identities of clusters based on shared gene expression patterns. Since the Allen institute data was generated using the SmartSeq2 protocol, Seurat's integrated anchor feature[77] was used to align the two datasets and transfer cell-type labels (Fig. S1e, f, S2e, f).

**Integrated analysis with spatial transcriptomics data to identify cell-types.** P7 HIPP single-cell data was used as a reference to transfer labels onto P8 spatial transcriptomics data in the barcoded region corresponding to the hippocampus using Seurat's integrated anchor feature[78] using default parameters.

**Single-cell trajectory analysis.** The velocyto python package (v 0.17.17)[79] was used to obtain.loom files from both replicates of HIPP and PFC single-cell data. After importing the UMAP co-ordinates of the datasets, the scVelo[79] package (v 0.2.0) and tutorial with default parameters were followed to acquire velocity plots (Fig. S1c, d, S2c, d). The cells involved in neurogenesis and neuronal differentiation in the dentate gyrus and hippocampus were subsetted based on cell-type identity, and slingshot trajectory analysis[45] was conducted on its first two principal components in an unsupervised manner (Fig. 5a).

**Alignment of Pacific Biosciences long read data.** Long read CCS fastqs sequences with PacBio were mapped and aligned to the reference genome (mm10) using STARlong and the following parameters:
-readFilesCommand zcat -runMode alignReads -outSAMattributes NH HI NM
  MD -readNameSeparator space --outFilterMultimapScoreRange 1
-outFilterMismatchNmax 2000 -scoreGapNoncan -20 -scoreGapGCAG -4
-scoreGapATAC -8 -scoreDelOpen -1 -scoreDelBase -1 -scoreInsOpen -1
-scoreInsBase -1 -alignEndsType Local -seedSearchStartLmax 50
-seedPerReadNmax 100000 -seedPerWindowNmax 1000
-alignTranscriptsPerReadNmax 100000 -alignTranscriptsPerWindowNmax 10000

**Alignment of spatial transcriptomics data sequenced using Oxford Nanopore.** Long reads sequenced on the ONT PromethION were mapped and aligned using minimap2 using the following parameters: -t 20 -ax splice -secondary=no

**Filtering of long reads for full-length, spliced, barcoded reads.** This was first described in our previous publication[80] but we outline the process here as well. We first filtered long reads by retaining only alignments with the following procedure. If the read had exactly one alignment and the mapping quality of that alignment was at least 20 (in the sam format mapping quality definition), this alignment was kept. If the read has multiple alignments, but one alignment's mapping quality outscored the maximum of all other mapping qualities for that read by at least 20, the highest scoring alignment was retained. Afterwards, we removed alignments that overlapped annotated ribosomal RNA genes. For spliced alignments, we determined the splice site consensus for each intron in the alignment and only retained, alignments, for which every single intron respected the GT-AG, GC-AG, or ATAC consensus. These alignments we have previously referred to as a consensus-split-mapped molecule (CSMM). For each CSMM, we counted the number of splice sites it shared with each annotated gene. The vast majority of CSMMs share splice sites with exactly one gene. Each CSMM was assigned to the gene with which it shared most splice sites (or all such genes, in the case of a tie). Additionally, we filtered for complete reads using published CAGE peaks and polyA site data[80,81] by only keeping CSMMs that had start and end sites falling within 50 bp of an annotated CAGE or polyA peak.

**Exon count assignment per cell-type.** High confidence mapped and aligned reads were processed sequentially and compared to the Gencode M21 gene annotation using an in-house script. A Gencode-annotated exon (Fig. S14a) was considered as being included in the read if both splice junctions were detected by the alignment. Due to the high error rate of ONT sequencing data, a variation of 3 bp was allowed for each splice junction, whereas for PacBio variation allowed was 2 bp, and overlapping exons were flagged (Fig. S14b). In cases where a read spanned an exon, but its splice junctions were not detected, the exon was considered excluded (Fig. S14c). Although ONT reads are known to often represent truncated versions of transcripts, terminal exons were counted but were only considered included when covered completely. Partially covered terminal exons were considered neither included nor excluded, and were discarded from the analysis (Fig. S14d). Further, exon counts were aggregated to produce exclusion and inclusion rates for each particular cell group.

**Differential isoform tests.** For isoform tests, a read was represented by a string denoting the TSS, introns, and polyA-site and each isoform was assigned an ID, with lower numbers corresponding to higher abundances. Next, counts for each isoform ID were assigned to individual cell-types. For each differential abundance test between two categories, genes were filtered out as 'untestable' if reads did not reach sufficient depth (25 reads/gene category). For genes with sufficient depth, a maximum of an $11 \times 2$ matrix of counts denoting isoform×category was constructed with the first ten rows corresponding to the first ten isoforms, and the last row comprised of collapsed counts from all the other isoforms (if any). P-values from a $\chi^2$ test were reported per gene, along with a ΔΠ value per gene. The ΔΠ was constructed as the sum of change in percent isoform (Π) of the top two isoforms in either positive or negative direction. After these numbers were reported for all testable genes for a comparison, the Benjamini Hochberg (BH)[82] correction for multiple testing with a false discovery rate of 5% was applied to return a

corrected p-value. If this FDR p-value was $< = 0.05$ and the $\Delta\Pi$ was more than 0.1 in one direction, i.e., the change in percent inclusion of one or two isoforms was more than 10% from one category to the other, then the gene was considered to be significantly differentially spliced.

**Differential TSS tests**. For each read, we determined the TSS as described above. The counts of each TSS in two conditions (e.g., excitatory and inhibitory neurons) were then summarized in a $n \times 2$ table of counts. As with the isoform tests, this table was reduced to a $11 \times 2$ table where the first ten rows represented the most abundant TSS and the 11th represented all the other TSS. Testing was performed as described above for isoforms.

**Differential polyA-site test**. For each read, we determined the polyA-site as described above. The procedure for testing was the same as described above for differential TSS.

**Differential exon tests**. Testing for differential exon inclusion followed a separate framework. For testing differential inclusion between two categories, a $2 \times 2$ table of inclusion and exclusion counts per exon was constructed if it was not constitutively spliced ($0.1 <= \Psi <= 0.9$) without considering counts in individual categories. An exon was considered for testing if the expected counts followed the criteria[83,84] $\frac{\min(rowSums) \times \min(colSums)}{Total} \geq 5$, and the p-value from a $\chi^2$ test was reported, along with a $\Delta\Psi$ value. The $\Delta\Psi$ was constructed as the difference in percent spliced in (PSI/$\Psi$) across the two categories. After these numbers were reported for all testable exons for a comparison, since these tests are dependent, the Benjamini Yekutieli (BY)[85] correction for multiple testing with a false discovery rate of 5% was applied to return a corrected p-value. If this p-value was $< = 0.05$ and the $\Delta\Psi$ was more than 0.1 i.e., the change in $\Psi$ was more than 10% between the two categories, then the exon is considered to be significantly differentially included.

**Isolating non-overlapping exons from single-cell data for correlation with spatial ONT data**. To confirm region-specific DIE in a technologically orthogonal fashion, we used spatial data sequenced with Oxford Nanopore as opposed to single-cell data sequenced with PacBio. Because of the relatively high error rate, truncated transcripts (~200 bp difference in average length of ONT vs PacBio), and mis-mapping of overlapping splice sites, we chose to work with exons instead of full-length transcripts. We isolated exons from the single-cell PacBio data that showed region-specific DIE without p-value correction and extracted corresponding exons from the spatial data if they satisfied the following condition. If an exon was reported to have alternative donor and acceptor sites, it was only accepted if more than 90% of reads overlapping the exon mapped to that particular exon. We then used this list of exons to calculate region-specific DIE using spatial exon expression and correlated the $\Delta\Psi$ values with the $\Delta\Psi$ values from the single-cell data. Despite high technical variation, we got high concordance (85% change in the same direction, $n = 40$).

**Building an enhanced annotation**. We isolated polyadenylated, barcoded, and spliced long-reads,

- whose TSS were within 50 bps of a published CAGE-peak[80]
- whose mapped read-end fell within 50 bps of a published polyA-site[81]
- whose intron-chains were inconsistent with any annotated transcript or any truncated version of an annotated transcript[86,87]
- whose internal exons (meaning both splice sites) were each supported by two or more single short reads (with ≥2 splicing events) or paired-end Illumina read pairs from a bulk-sequencing experiment
- whose introns were each supported by two or more spliced Illumina reads from a bulk sequencing experiment
- who could not be interpreted as a truncated version of another such novel isoform

This resulted in an enhanced annotation, in which all added isoforms had a unique, previously absent intron-chain.

These isoforms as well as already annotated isoforms received read counts in each cell types according to the single-cell long-read dataset in each brain region. Novel isoforms intron-chains are by construction (see above) unique; however, annotated isoforms can differ only in their TSS or polyA-site. A long-read was therefore assigned to a GENCODE transcript that minimized the sum of abs (readEndMapping—annotatedIsoformEnd) and abs(readStartMapping—annotatedIsoformStart). In the case of a tie, only the divergence at the TSS was considered.

This annotation thus includes counts for thousands of isoforms in different cell types, such as "P7Hipp_OPCs" (representing oligodendrocyte precursor cells from a hippocampus at postnatal day 7) and "P7PFC_OPCs" (representing oligodendrocyte precursor cells from a pre-frontal cortex at postnatal day 7).

**Manual validation from the GENCODE team**. Novel isoforms were manually verified and integrated into the GENCODE geneset[88] in accordance with guidelines developed by the HAVANA group for the GENCODE / ENCODE projects[89].

Novel introns not previously identified within the GENCODE geneset were independently confirmed by intron predictions taken from the Ensembl RNAseq pipeline using ENCODE data[90,91], whilst novel transcription start sites were validated based on the Cap Analysis of Gene Expression (CAGE) libraries generated by FANTOM[92]. Finally, each new transcript model was assigned a 'biotype' indicative of its presumed functional categorization, with the biotypes matching to those used in GENCODE as described by Frankish et al.[88].

**Validation of cell-type-specific splicing using Basecope Duplex Detection Assay**. Target transcriptomic regions for individual isoforms were isolated from the DIE analysis and probes were designed by Advanced Cell Diagnostics (ACD). Postnatal day 7 mice were transcardially perfused, and 12 μm brain sections were processed according to the manufacturers recommendation for BaseScope Duplex Detection Assay (Protocol number 323800-USM, Advanced Cell Diagnostics, ACD) with a pre-treatment according to the protocol for fixed, frozen tissue (Protocol number 320534, Advanced Cell Diagnostics, ACD) with only 15 min of Protease Plus incubation. Images were taken using a Leica DM5500 B with a Leica DFC295 camera.

**Reporting summary**. Further information on research design is available in the Nature Research Reporting Summary linked to this article.

## Data availability

All data used for this study is publicly available on GEO under the accession token GSE158450. All data supporting the findings of this study are provided within the paper and its supplementary information. All additional information will be made available upon reasonable request to the authors.

## Code availability

The source code for the methods and visualizations are available as functions in an R-package[93] (https://github.com/noush-joglekar/scisorseqr) and processed single-cell and spatial data is available for download at www.isoformAtlas.com

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

## Acknowledgements

We thank Jenny Xiang, Dong Xu, and Adrian Tan at the Genomics Resources Core facility at Weill Cornell Medicine for help with facilitating sequencing, and Ishaan Gupta for helping with initial processing of the samples. We also thank Christopher Mason and Daniel Butler for use of their PromethION machine. This work was supported by the Brain Initiative (grant 1RF1MH121267-01 to H.U.T), NIGMS (grant 1R01GM135247-01 to H.U.T), NINDS (grant 1R01NS105477 to M.E.R), NIDA (T32DA03980 to S.L.), and NIMH (grant R01MH118934 to G.P.), Australian NHMRC Early Career Fellowship (APP1156531 to S.A.H), RFBR (grant 19-04-01074 to A.P.), D.R. was supported by Programma per Giovani Ricercatori Rita Levi Montalcini granted by the Italian Ministry of Education, University, and Research, by the Chan Zuckerberg Initiative DAF, an advised fund of Silicon Valley Community Foundation (CZF2019-002443), and by the NCI (2U24CA180996), S.A.S was supported by the Brain and Behavior Foundation NARSAD YIA and Sontag Foundation, P.F. was supported by NHGRI (grant 2U41HG007234), Wellcome (WT108749/Z/15/Z), and the European Molecular Biology Laboratory. E.D.J was supported by HHMI.

## Author contributions

A.J. and H.U.T. devised the experiments. P.G.C., S.L., A.K.S., B.H., O.F., M.Y.W., W.L., J.M., J.G.C., A.H., and N.I.W. performed the experiments. A.J., A.M., A.P., D.R., S.R.W., A.F., and H.U.T devised the analyses. A.J., A.P., A.M., Q.W., T.H., S.A.H., A.S., and H.U.T. performed the analyses. All of the authors discussed and interpreted the results throughout the project. A.J. and H.U.T. wrote the paper with inputs from all of the other authors. C.D., Z.B., D.R., E.D.J., O.F., S.A.S., A.F., P.F., W.L., G.S.P., A.B.S, M.E.R., and H.U.T. supervised the project.

## Competing interests

J.G.C., A.H., N.I.W., S.R.W., and Z.B. are employees of 10X Genomics. PF is a member of the scientific advisory boards of Fabric Genomics, Inc., and Eagle Genomics, Ltd. A.J., A.P., A.M., P.G.C., S.L., A.K.S., J.M., B.H., M.Y.W., A.N.S., S.A.H., T.H., Q.W., C.D., O.F., S.A.S., D.R., E.D.J., W.L., G.P., A.F., A.B.S., M.E.R., and H.U.T. declare no competing interests.
