## [Peer Review File · Nature Communications]

Reviewers' Comments:

Reviewer #1:

Remarks to the Author:

The manuscript by Joglekar et al presents a systematic study of alternative transcript isoform usage in hippocampus and prefrontal cortex of mouse brain. They use long-read sequencing technology on RNA isolated from single cells (10x chromium), and from spatially barcoded arrays (10x visium). They observed usage of different isoforms for 395 genes, and further dissect these differences to celltypes. As a technology developer I cannot judge the impact of these observations, but have focused my attention on the technical aspects of this work. The authors have provided a very detailed description of the experimental protocols, and the computational processing of the data, as well as the statistics applied, and I have no reasons to doubt the validity of the conclusions drawn. I found the manuscript over all well written and easy to follow. I have only one question and one comment:

The authors applied a threshold of isoform usage of 0.1 to be considered differentially used. My question is how to envision the functional implication of cells expressing 10% of one variant and 90% of the other. How could the 10% contribution impact a cell's overall phenotype? Or does this number reflect a minority population of cells that predominantly express the variant isoform? I think this could need some further discussion.

Figure 2, 3, 5: Some text and graphical details are too small to read.

Reviewer #2:

Remarks to the Author:

In "Cell-type, single-cell, and spatial signatures of brain-region specific splicing in postnatal development Joglekar et al. describe a novel approach to quantify differential expression of isoforms between cell types in hippocampus and prefrontal cortex.

The approach is interesting and is an important contribution to understand functional diversity in cell types and between brain regions.

I will appreciate if the author will answer at my questions.

Page 5: The authors use "non-neuronal glia" and "glial non-neurons " in the text to define astrocyte, oligos and microglia. This is redundant because by definition glia cells are not neurons.

Page 5: The dissection of the contribution of microenvironment versus origin is very interesting. If it is possible to determine the contribution of the different genes to the Concurrent regional DIE, Do you think that any of the genes selected using short reads to define cell types will play any type of role in affecting the concurrency?

Page 5: "...Interestingly, comparisons between non-neuronal cell types showed higher DIE than those observed within neurons". Do you think that this difference has to do with the embryonical origin of the type of cell or because the non-neuronal cells are covering a larger spectrum of functions compared to neurons that even though different, have a more similar function? Example comparing oligodendrocyte and endothelial cells.

Page 8: "Our results indicate that understanding the cell-type basis of sample-specific DIE requires a thorough understanding of cell-type specific DIE within each sample. " It is not clear to me what you consider as sample, does it mean tissue-specific or it refers to variations between samples from the same tissue?

Reviewer #3:

Remarks to the Author:

In this manuscript, Joglekar and co-workers present an investigation of differential isoform usage across two post-natal brain regions and cell types therein. For this purpose, they use SciSOSeq, a hybrid approach that allows to define single cell identities using 3' seq and isoform quantifications using long reads, which are then pooled based on the cell type identity. In this regard, it is important to note that the analysis is not based on single cells, but on single cell types.

The authors report a series of findings of interest. For instance, they find that, in most cases, differences in isoform usage between brain regions is due to changes in splicing between only one cell type, something that, to my knowledge, has not been directly shown (or even addressed) before. Other findings are more descriptive, on individual exons and genes. Overall, I believe this is an important step forward on the technological front, and I would support its publication after some relatively minor revisions.

Some comments:

1) The authors should be careful to avoid overstating the novelty of their study. It is indeed novel from a technical point of view, but statements such as "Our single-cell isoform data enable, for the first time, the illumination of alternative splicing regulation across cell types" (Discussion) are by no means correct. Multiple studies before have addressed this issue (even at higher depth and level of detail) using bulk Illumina sequencing of carefully isolated cell types (e.g. Furlanis et al, Nature Neurosci 2019). While the use of single-cell-derived long-read-based data is novel, many of the questions and conclusions are not. Therefore, I suggest a careful revision of the text to avoid misleading statements.

2) The authors report that their isoform-level tests are more sensitive than exon-level tests. However, if I understood it correctly, I do not think this is a fair comparison. In their exon-level tests, they required a delta psi higher than 10, whereas in the isoform-level tests they require a ****combined**** delta usage of 10 (from at most two isoforms). Thus, this seems equivalent similar to a delta psi of 5 for exons, if the inclusion and exclusion (1-psi) are considered as separate isoforms. Consistent with this interpretation, their median delta usage for individual DIE isoforms is in fact ~0.8-0.9 (Fig 2e). How does the test look if they require a delta usage of 10 for at least one individual isoform? What if they require a minimum delta psi of 5 for the exon level analyses? Please clarify.

3) In Fig 3, they show that brain region identity can override cell type identity in terms of isoform usage. They provide two alternative explanations and seem to favor one of them (microenvironment effects). I wonder, however: could this be due to experimental differences? I assume the two brain regions were processed at least in part separately, which could cause some "regional" differences affecting all their cell types (e.g. biases due to RNA degradation, etc.). To rule out this possibility, I suggest the authors test: (i) if the same regional differences are observed in both replicates (i.e. the overlap of regional changes at the isoform level); and (ii) if there are specific biases in these exons (e.g. do they tend to be in the 3' or 5' of transcripts?).

4) I could not find how the authors defined the background for the GO enrichment analysis of Fig 5. Given the sparse nature of their data, this is an important point to avoid expected enrichments due to gene expression biases.

5) I am a bit puzzled about the results on the subcellular localization of Fgf13. As far as I know, FGfs are secreted morphogens that act by activating their receptors (FGFRs) in the target cells, thereby regulating multiple processes. How do they propose the two isoforms perform their differential roles in nucleolus and cytoplasm? Does Fgf13 act differently to other members of the family? Please clarify. The co-staining experiments are also not very obvious to interpret (at least,

I could see no obvious co-staining).

6) The manuscript often comes across as a collection of small observations on individual exons and genes whose splicing patterns are relatively well known. I recommend the authors avoid this whenever possible and focus only on the most novel and interesting bits that could not be captured by other approaches. In fact, most of these results are only in supplementary figures, and I wonder they should be there at all. Perhaps they could have a supplementary figure with all these descriptions for experts interested in the processing of individual genes.

Cell-type, single-cell, and spatial signatures of brain-region specific splicing in postnatal development

Joglekar A. et al

Reviewer #1 (Remarks to the Author):

The manuscript by Joglekar et al presents a systematic study of alternative transcript isoform usage in hippocampus and prefrontal cortex of mouse brain. They use long-read sequencing technology on RNA isolated from single cells (10x chromium), and from spatially barcoded arrays (10x visium). They observed usage of different isoforms for 395 genes, and further dissect these differences to celltypes. As a technology developer I cannot judge the impact of these observations, but have focused my attention on the technical aspects of this work. The authors have provided a very detailed description of the experimental protocols, and the computational processing of the data, as well as the statistics applied, and I have no reasons to doubt the validity of the conclusions drawn. I found the manuscript over all well written and easy to follow. I have only one question and one comment:

We thank Reviewer #1 for their summary and encouraging words about our manuscript

The authors applied a threshold of isoform usage of 0.1 to be considered differentially used. My question is how to envision the functional implication of cells expressing 10% of one variant and 90% of the other. How could the 10% contribution impact a cell's overall phenotype? Or does this number reflect a minority population of cells that predominantly express the variant isoform? I think this could need some further discussion.

The reviewer raises an important question. Until single-cell technologies became viable, it was practically impossible to make statements as to whether

- a small fraction of cells express the minor isoform while every other cell expresses the major isoform
- each cell predominantly expresses the major isoform, and the minor isoforms in smaller abundances

This question has been addressed in cell-lines (e.g. Song et al, 2017). For our dataset, the depth of sequencing per *gene X cell* combination is too low for us to attempt to answer this on a transcriptome wide scale, or make claims about the phenotypic consequences of DIE. However, to answer the question of whether individual cells reflect the isoform choices of cell types, we isolated 59 genes for which we had at least 50 cells wherein we see at least 8 reads. Of those, there is only 1 gene – *Nnat* which fulfills the following:

- have at least 8 reads per cell
- represented in over 3 or more cell-types
- alternatively spliced ($0.1 \leq \Pi \leq 0.9$ for two isoforms)
- exhibit strong switch-like isoform change between cell types

For *Nnat*, we calculated the percent inclusion (π) of its two main isoforms per individual cell and represented them in two violin plots (one for each isoform) for each cell type. In ependymal cells, the “left” isoform (represented by the left boxplot) is used in the overwhelming majority of molecules in all cells. In excitatory neurons however, almost all cells use the isoform on the right. Thus, a difference between the two cell-types is represented in close to all individual cells. Granule neuroblasts (GranuleNB) on the other hand, have huge variability in isoform abundance. Differences between isoform abundances between granuleNB and excitatory neurons are not represented by all cells in the two populations. Thus, both models appear to exist for *Nnat*. We hope to address validity of this observation for thousands of genes with more detailed sequencing technologies in the future.

We have added the above figure as Fig 4f to explain this and have added the following text to the manuscript

“Single-cell basis of DIE between cell types

When DIE is observed between two cell types, two competing hypotheses can explain this phenomenon⁴. Either all cells of each cell type behave uniformly and reflect the differences in isoform expression between the two cell types, or there is variability in isoform expression among individual cells of one or both cell type(s). Neuronatin (*Nnat*) is an important developmental gene expressing a neuron-specific isoform. In *Nnat*, DIE between ependymal cells and excitatory neurons is represented by the vast majority of individual cells. However, the case of DIE between excitatory neurons and granule neuroblasts is different: some granule neuroblasts behave like excitatory neurons, while others behave like non-neurons. This may be due to different sub-populations of granule neuroblasts (Fig 4f)“

Figure 2, 3, 5: Some text and graphical details are too small to read.

We apologize for the difficulty in interpreting the figures. We had resized the original figures so as to fit the graphics and legend onto one page, without realizing the extent to which this would inconvenience the reader. We have now included the figures in their original size, increased the text size of axis labels, and moved the legend to the next page.

Reviewer #2 (Remarks to the Author):

In “Cell-type, single-cell, and spatial signatures of brain-region specific splicing in postnatal development Joglekar et al. describe a novel approach to quantify differential expression of isoforms between cell types in hippocampus and prefrontal cortex.

The approach is interesting and is an important contribution to understand functional diversity in cell types and between brain regions.

I will appreciate if the author will answer at my questions.

We thank Reviewer #2 for their summary and attempt to carefully answer each point raised by them below:

Page 5: The authors use “non-neuronal glia” and “glial non-neurons” in the text to define astrocyte, oligos and microglia. This is redundant because by definition glia cells are not neurons.

We had originally used the distinction of non-neuronal glia and non-neuronal non-glia to separate comparisons involving astrocytes, oligodendrocytes, and microglia from vascular cells. However, as Reviewer #2 pointed out, this distinction is needlessly complicated, and we have now amended the text to reflect the changes to “glia” when referring to astrocytes and oligodendrocytes, and “vascular + immune” when referring to endothelial cells, pericytes, microglia, and macrophages

Page 5: The dissection of the contribution of microenvironment versus origin is very interesting. If it is possible to determine the contribution of the different genes to the Concurrent regional DIE, Do you think that any of the genes selected using short reads to define cell types will play any type of role in affecting the concurrency?

We hope that we interpret this question correctly. Specifically, we interpret it in two ways:

Can cell-type markers influence concurrent DIE between brain regions across multiple cell types?

The expression of splicing factors like *NeuN* that are selectively expressed in neuronal cell types affects the splicing of various genes only in neurons, and hence works against the above concurrence. However, knowing which splicing factor affects which isoform is currently difficult to measure from our data, so that we cannot provide a conclusive answer with these data alone.

Do genes with concurrent DIE between brain regions across multiple cell types influence cell type definition?

We can indeed determine the genes that significantly contribute to the concurrent regional DIE, however, we must point out that in order to deem a gene testable, it has to be sufficiently expressed in at least two cell types. Therefore, we expect genes with concurrent DIE in multiple cell types between brain regions to be less variable in gene expression than other genes. Indeed, we observe that only 3 out of 2000 variable genes show concurrent DIE between brain regions across neurons and non-neurons.

If we exclude these three genes from single-cell clustering analysis, we find only very slight alterations in cell type definition. The only drastic difference is that the hippocampal InhibNeuron-2 cluster breaks up into 2 clusters. The heatmaps below represents the Jaccard index (intersection/union of cells) between any two clusters in a pairwise fashion.

Page 5: "...Interestingly, comparisons between non-neuronal cell types showed higher DIE than those observed within neurons". Do you think that this difference has to do with the embryonal origin of the type of cell or because the non-neuronal cells are covering a larger spectrum of functions compared to neurons that even though different, have a more similar function? Example comparing oligodendrocyte and endothelial cells.

Reviewer #2 raises an important question. We completely agree with them in that there are two competing, but non-mutually exclusive explanations that relate to the abundance of DIE between two cell types.

1. How functionally diverse are the two cell types?
2. Do the two cell types arise from the same stem cell?

Overall, we cannot answer the question satisfactorily because we don't have a good metric that defines how functionally similar two cell types are. However, we can find examples of model 1 overriding model 2 and vice-versa.

If we consider the comparison between excitatory neurons and astrocytes, which has a very high DIE fraction (35.8% of tested genes have DIE, Fig 4a). Both cell-types originate from RGLs but have clearly distinct functions, and therefore this serves as an example of functional differences driving splicing diversity.

On the other hand, CA1-CA3 pyramidal neurons (ExcitNeuron1) are somewhat functionally similar to Cajal-Retzius cells (InhibNeuron3) in that they are both neurons, but do not originate from the same stem cells. The two groups exhibit fairly high (18%) DIE, which would indicate that different embryonic origin overrides functional similarity in driving splicing diversity.

We have represented this in the discussion as follows:

“However, we find that non-neuronal cell types exhibit high pairwise DIE. This may be partially due to the functional diversity of non-neuronal cells, or due to origins from different stem cell populations. Importantly, excitatory neurons and astrocytes which originate from the same stem cell population but are functionally distinct show very high DIE (35.8%). However, the excitatory neuron population also shows high DIE with Cajal-Retzius cells (18%) despite both originating from different stem cell populations but functioning as neurons. Among the non-neurons, choroid plexus epithelial cells show particularly large differences from other non-neuronal (but also neuronal) cell types.”

Page 8: “Our results indicate that understanding the cell-type basis of sample-specific DIE requires a thorough understanding of cell-type specific DIE within each sample.” It is not clear to me what you consider as sample, does it mean tissue-specific or it refers to variations between samples from the same tissue?

We apologize for the lapses in our text. We should not have employed the word ‘sample’ since it has multiple interpretations. We now use ‘brain region’ in this context, and have made the following changes in the text to clarify the statement.

“Our results indicate that understanding the cell-type basis of brain-region specific DIE requires a thorough understanding of cell-type specific DIE within each brain region.”

Reviewer #3 (Remarks to the Author):

In this manuscript, Joglekar and co-workers present an investigation of differential isoform usage across two post-natal brain regions and cell types therein. For this purpose, they use SciSORseq, a hybrid approach that allows to define single cell identities using 3' seq and isoform quantifications using long reads, which are then pooled based on the cell type identity. In this regard, it is important to note that the analysis is not based on single cells, but on single cell types.

The authors report a series of findings of interest. For instance, they find that, in most cases, differences in isoform usage between brain regions is due to changes in splicing between only one cell type, something that, to my knowledge, has not been directly shown (or even addressed) before. Other findings are more descriptive, on individual exons and genes. Overall, I believe this is an important step forward on the technological front, and I would support its publication after some relatively minor revisions.

We thank Reviewer #3 for their summary and encouraging words about our manuscript and attempt to address each of their comments below

Some comments:

1) The authors should be careful to avoid overstating the novelty of their study. It is indeed novel from a technical point of view, but statements such as "Our single-cell isoform data enable, for the first time, the illumination of alternative splicing regulation across cell types" (Discussion) are by no means correct. Multiple studies before have addressed this issue (even at higher depth and level of detail) using bulk Illumina sequencing of carefully isolated cell types (e.g. Furlanis et al, Nature Neurosci 2019). While the use of single-cell-derived long-read-based data is novel, many of the questions and conclusions are not. Therefore, I suggest a careful revision of the text to avoid misleading statements.

The reviewer is absolutely correct, and we apologize for the phrasing. We now write:

"Building on the short-read investigation of cell-type specific alternative splicing in the brain⁷², our data enable the illumination of full-length isoform regulation across cell types."

2) The authors report that their isoform-level tests are more sensitive than exon-level tests. However, if I understood it correctly, I do not think this is a fair comparison. In their exon-level tests, they required a delta psi higher than 10, whereas in the isoform-level tests they require a ****combined**** delta usage of 10 (from at most two isoforms). Thus, this seems equivalent similar to a delta psi of 5 for exons, if the inclusion and exclusion (1-psi) are considered as separate isoforms. Consistent with this interpretation, their median delta usage for individual DIE isoforms is in fact ~0.8-0.9 (Fig 2e). How does the test look if they require a delta usage of 10 for at least one individual isoform? What if they require a minimum delta psi of 5 for the exon level analyses? Please clarify.

We apologize for failing to explain the details of our procedure, our text was lacking crucial information. For the gene-wise $\Delta\Pi$ calculation, we employ at most two isoforms. Two isoforms are considered only if their isoform $\Delta\Pi$ point **in the same direction**. The motivation for considering more than one isoform contributing to the $\Delta\Pi$ is as follows: if there are three or more isoforms, the change of a single exon can

be distributed across two or more isoforms and considering only one isoform is a very conservative approach. On the other hand, considering no more than two isoforms avoids summing lots of tiny changes to achieve a large delta. A drawback to our approach are cases like *Dscam* in *Drosophila*, where a single exon may be part of thousands of isoforms. In such cases, summing over multiple isoforms or performing exon-tests is more advantageous

Thus, in the situation of a gene with a single alternative exon (and thus two isoforms), where we have a $\Delta\Pi = x$ for the inclusion isoform and a $\Delta\Pi = -x$ for the exclusion isoform, for the gene-wise $\Delta\Pi$, only one isoform is considered.

We have now modified the text to emphasize the directionality as follows:

“Similarly to requiring a $\Delta\Psi \geq 0.1$ for short reads², we require $FDR \leq 0.05$ and $\Delta\Pi \geq 0.1$. This $\Delta\Pi \geq 0.1$ can be contributed collectively by at most two isoforms, provided their isoform $\Delta\Pi$ point in the same direction, to consider a gene exhibiting differential isoform expression (DIE, Fig 2a).”

That being said, we calculated the number of significant genes by the exon tests when considering a 5% $\Delta\Psi$, and the number increased from 31 to 36 genes with significant exons. For the isoform tests, if we impose the more rigorous cutoff of 10% change contributed by at most one isoform, then we lose 107 out of 395 significant genes, leading to 288 genes showing significant regional DIE. This indicates that regardless of the cutoffs imposed, gene-based tests are more sensitive than exon-based tests at detecting DIE.

Intersection of significant genes by exon-based and isoform-based tests

3) In Fig 3, they show that brain region identity can override cell type identity in terms of isoform usage. They provide two alternative explanations and seem to favor one of them (microenvironment effects). I wonder, however: could this be due to experimental differences? I assume the two brain regions were processed at least in part separately, which could cause some "regional" differences affecting all their cell types (e.g. biases due to RNA degradation, etc.). To rule out this possibility, I suggest the authors test: (i) if the same regional differences are observed in both replicates (i.e. the overlap of regional changes at the isoform level); and (ii) if there are specific biases in these exons (e.g. do they tend to be in the 3' or 5' of transcripts?).

We thank the reviewer for this comment. The question of whether this result is replicable is important. Briefly, the answer is yes, it is replicable when using both replicates. In detail, we proceeded as follows: In the original analysis using Replicate 1 only, we asked what fraction of genes had a $\Delta\Pi$ of at least 0.1 in neurons (fraction f_1), and in non-neurons (fraction f_2). Assuming independence, the random expectation of co-occurrence is $f_1 \times f_2$, which we compared to the observed co-occurrence of $\Delta\Pi \geq 0.1$ in neurons and in non-neurons. This observed occurrence was significantly greater than random expectation, which we showed in Fig 3E.

We have now re-done this analysis, taking into account both replicates, and denoted the fraction of genes having $\Delta\Pi \geq 0.1$ in neurons in both replicates as F_1 , and likewise F_2 in non-neurons. We then repeated the above analysis and found that the observed co-occurrence to be significantly greater than random expectation, i.e. $F_1 \times F_2$

Note that when using both replicates, the expected co-occurrence is much lower than when using only one replicate (i.e. $F_1 \times F_2 \ll f_1 \times f_2$) simply because $F_1 < f_1$ and $F_2 < f_2$. F_1 is lower than f_1

because we have an additional criterion to be fulfilled by F1, i.e. the value has to be observed in both replicates as opposed to only one replicate for f1.

We have added the analysis based on both replicates as a supplemental figure (S7d) and the following text to the manuscript:

“Not only do we see concurrence in Rep1, but we find that the observation is conserved in all investigated levels across both replicates considered together (Fig S7d)”

4) I could not find how the authors defined the background for the GO enrichment analysis of Fig 5. Given the sparse nature of their data, this is an important point to avoid expected enrichments due to gene expression biases.

Reviewer #3 raises a very important point. Initially we had done this analysis in two ways - once with a background of the whole transcriptome, and once with a background of all tested genes which are highly expressed in our dataset. These gave similar results, in that both find spliceosomal alterations in the transition of RGL to NIPCs, and synaptic changes in the Granule Neuroblasts to Excitatory Neuron transition. However, using the tested genes as a background set did not show splicing changes in the transition from GranuleNB to excitatory neurons

Unfortunately, in Fig 5 we presented the one with the whole transcriptome as a background. We agree that this is more subject to bias and have now replaced it with the analysis using the tested genes (1k-2k genes depending on the condition) as a background. This changes the figure (Fig 5b-d, S10-S11), but the only change to the text is the removal of the word “additionally” in the following context (as splicing is no longer significant in one of the comparisons):

“However, as granule neuroblasts matured to excitatory neurons, DIE was associated with synapse formation and axon elongation (*Snap25*, *Snca*, *Syp*, *Dbn1*, *Cdc42*, *Nptn*, *Gap43*) among others”

In summary, we thank the reviewer for pointing out this shortcoming.

5) I am a bit puzzled about the results on the subcellular localization of Fgf13. As far as I know, FGFs are secreted morphogens that act by activating their receptors (FGFRs) in the target cells, thereby regulating multiple processes. How do they propose the two isoforms perform their differential roles in nucleolus and cytoplasm? Does Fgf13 act differently to other members of the family? Please clarify. The co-staining experiments are also not very obvious to interpret (at least, I could see no obvious co-staining).

The reviewer is completely right about FGFs in general. However, *Fgf11-14* (including our *Fgf13*) are exceptions within the FGF family, which we did not sufficiently describe in our original text. We now detail this by writing:

“A member of the fibroblast growth factor (FGF) superfamily, *Fgf13* is one of four FGF family members (*Fgf11-14*) labeled fibroblast growth factor homologous factors, which—unlike most FGFs—do not have signal sequences, are not secreted, and function intracellularly⁵⁸. Among its intracellular roles are regulation of voltage-gated sodium channels^{59–61}, rRNA transcription⁶², and microtubule stabilization⁵⁵.”

Regarding the co-staining question, we suspect that the figure quality and size may have led to a misunderstanding. There are four panels representing in-situ hybridization using the Basescope method in our Fig 5g:

- Fgf13-S isoform co-stained with *Neurod6*: Pink (Fgf13-S) **co-localizes** almost perfectly with blue (*Neurod6*) thereby shifting to purple. Thus, Fgf13-S is expressed almost exclusively in *Neurod6*+ excitatory neurons
- Fgf13-VY isoform co-stained with *Neurod6*: Pink (Fgf13-VY) **does not co-localize** with blue (*Neurod6*). Thus, Fgf13-VY is expressed outside of *Neurod6*+ excitatory neurons
- Fgf13-S isoform co-stained with *Gad2*: There is ample pink (Fgf13-S) staining outside of *Gad2* (blue) staining. Thus Fgf13-S **does not co-localize** with *Gad2*+ interneurons
- Fgf13-VY isoform co-stained with *Gad2*: There is no separate pink (Fgf13-VY) visible because all Fgf13-VY occurrences co-occur with *Gad2* (blue). Thus Fgf13-VY **co-localizes** with *Gad2*+ interneurons

In summary, the top image demonstrates that the *Fgf13-S* isoform localizes with *Neurod6*, a marker for excitatory neurons while *Fgf13-VY* does not. The bottom image shows that the *Fgf13-VY* isoform localizes with the inhibitory neuron marker *Gad2* while *Fgf13-S* does not.

We added the following description to the text to avoid any misunderstanding:

“This was confirmed using Basescope analysis with probes designed for excitatory and inhibitory neuron marker genes, and separate probes for the S and VY isoforms. We find that the S isoform (pink) co-localizes with excitatory neurons (blue, *Neurod6*) and not inhibitory neurons (blue, *Gad2*), whereas the VY isoform (pink) co-localizes with the inhibitory neurons but not the excitatory neurons”

We have also amended the figure legend as follows:

Figure 5 - Relative isoform expression differences during development reflect dynamic changes in function

g. Basescope (in situ hybridization) images of Fgf13-S and Fgf13-VY isoform expression (pink stain) in the hippocampus, with simultaneous staining for i. excitatory neurons (marked by *Neurod6* - blue) and ii. inhibitory neurons (marked by *Gad2* - blue). Each image panel features enlargements of the dentate gyrus for the S isoform, and CA1 region for the VY isoform **h.** Subcellular localization of overexpressed GFP-tagged Fgf13-S isoform in nucleolus and GFP-tagged Fgf13-VY isoform in cytoplasm.

6) The manuscript often comes across as a collection of small observations on individual exons and genes whose splicing patterns are relatively well known. I recommend the authors avoid this whenever possible and focus only on the most novel and interesting bits that could not be captured by other approaches. In fact, most of these results are only in supplementary figures, and I wonder they should be there at all. Perhaps they could have a supplementary figure with all these descriptions for experts interested in the processing of individual genes.

We appreciate the reviewer's concern for the economy of the main article, and the comment suggesting that a heavy focus on examples can break the flow of the manuscript and divert attention, especially when the reader is more interested in genome-wide patterns. Our intention was to also capture the attention of the synapse community, but we acknowledge that may have over-emphasized the background of some of the synaptic examples.

Therefore, we have now shortened these descriptions, especially the known background, while still mentioning the novelty of their cell-type specific isoform expression. We think this improves the overall flow without entirely discarding the observations, because the fact that these examples corroborate previous insights is important support for our results. We hope that the reviewer agrees to the usefulness of the improved approach.

For example, we have now reduced the descriptions for vesicle mediated endocytosis (*Snap25*, *Clta*, *Cltb*) from 250 words down to 180, and from 134 to 65 in the neurodevelopmental gene examples (*Dlgap4*, *Nptn*).

Reviewers' Comments:

Reviewer #1:

Remarks to the Author:

The authors have addressed my points in the revised manuscript, and think it is now acceptable for publication.

Reviewer #2:

Remarks to the Author:

I am satisfied with the additional work added by the authors to the manuscript.

Reviewer #3:

Remarks to the Author:

The authors have successfully addressed all my concerns. Congratulations for the great work.

Cell-type, single-cell, and spatial signatures of brain-region specific splicing in postnatal development

Joglekar A. et al

Reviewer #1 (Remarks to the Author):

The authors have addressed my points in the revised manuscript, and think it is now acceptable for publication

Reviewer #2 (Remarks to the Author):

I am satisfied with the additional work added by the authors to the manuscript.

Reviewer #3 (Remarks to the Author):

The authors have successfully addressed all my concerns. Congratulations for the great work.

We thank all the reviewers for their insightful comments, time, and effort in the review process – the manuscript benefited greatly. We recognize that the reviewers made extra effort to be quick in their replies and we really appreciate that.